



# Characterization of event water fractions and transit times under typhoon rainstorms in fractured mountainous catchments: Implications for time-variant parameterization

Jun-Yi Lee[1], Yu-Ting Shih[1], Chiao-Ying Lan[1], Tsung-Yu Lee[2], Tsung-Ren Peng[3], Cheing-Tung Lee[1], Jr-Chuan Huang[1*]

[1]Department of Geography, National Taiwan University, Taipei City, 10617, Taiwan.
[2]Department of Geography, National Taiwan Normal University, Taipei City, Taiwan
[3]Department of Soil and Environmental Sciences, National Chung Hsing University, Taichung City, Taiwan.

*Correspondence to*: Jr-Chuan Huang (riverhuang@ntu.edu.tw)

**Abstract.** Transit time with its indicative significance in regulating rainfall-runoff mechanism is a key factor for understanding many biogeochemical processes, but is rarely investigated in steep and fractured mountainous catchments. Mountainous catchments in Taiwan are characterized by active endogenic tectonics and exogenic typhoons and thus provide opportunities to explore the hydrodynamic systems over time. In this study, the hydrometrics and $\delta^{18}O$ in rain and stream water were sampled by ~3-hour interval for six typhoon events in two mesoscale catchments. The TRANSEP (transfer function hydrograph separation model) and global sensitivity analysis was applied for estimating mean transit time ($MTT_{ew}$) and fraction ($F_{ew}$) of event water and identifying the chronosequent parameter sensitivity. Results show that TRANSEP could satisfactorily simulate the streamflow and $\delta^{18}O$ change with the efficiency coefficients of from 0.85 to 0.97 and from 0.61 to 0.99, respectively. The $MTT_{ew}$ and Few varied from 2 to 11h and from 0.2 to 0.8, respectively. Our $MTT_{ew}$ in the meso-scale catchments is similar with that in micro-scale catchments, showing a fast transfer in our steep catchments. The mean rainfall intensity which negatively controls on the $MTT_{ew}$ and positively on the $F_{ew}$ is a predominant indicator which likely activates preferential flow paths and quickly transfers event water to the stream. Sensitivity analysis among inter- and intra-events suggested that parameter sensitivity is event-depend and time-variant, affirming a nonlinear behavior in event water transfer function and time-variant parameterization should be particularly considered when estimating the $MTT_{ew}$ in steep and fractured catchments.

## 1 Introduction

Transit time chronicling the elapsed time of water from entering a system, traveling through the system to its exit of the system (Stewart and Mcdonnell, 1991) implies information about sources, pathways, and water storage. It also indicates how catchments retain and release water and solutes associated with biogeochemical processes (Stark and Stieglitz, 2000), chemical weathering (Maher and Chamberlain, 2014) and contamination transportation (Kirchner, 2003;Kirchner et al., 2000) Therefore, accurately estimating mean transit time could provide a physical basis for understanding the hydrological behaviors and consequently the biogeochemical effects of anthropogenic disturbances (Turner et al., 2006) and landcover change (Burns et





al., 2005;Buttle et al., 2001). However, the dynamics of transit time change during a rainstorm is rarely studied, which is important to understand the event driven pollution (McDonnell et al., 2010).

There are three approaches commonly used to estimate transit time; they are time domain convolution integral methods (e.g., Maloszewski and Zuber, 1982), spectral analysis (e.g., Kirchner et al., 2000) and the tracer-aided rainfall–runoff models
(e.g., Birkel and Soulsby, 2015). Each approach has its own advantages/disadvantages, whereas the tracer-aided rainfall-runoff models which can improve the understanding of storage dynamic, mixing degree and transit time within the catchment (Botter, 2012; Botter et al., 2010; Harman, 2015) are widely applied. The TRANSEP (transfer function hydrograph separation) proposed by Weiler et al. (2003) belonging to tracer-aided rainfall-runoff models can estimate the transit time at the event time-scale. Simply, TRANSEP model integrates the unit hydrograph and the isotope hydrograph separation (IHS) techniques
in a quantitative approach and partitions the temporal variability of streamflow into event and pre-event components.

Most studies on mean transit time of event water ($MTT_{ew}$) are lack environmental variety in terms of climate zone, site scale as well as precipitation magnitude. Almost all studies, 50 events that we reviewed in the literature, are in the temperate and few in tropical zones, but none in the Subtropical. Among all of the studies, the average total precipitation of an event is below 100 mm, ranging from 120 to 250 mm, except three events reported by McGuire and McDonnell (2010) and Segura et
al. (2012). The catchment area ranges from 0.07 to 31 km$^2$ and the catchment slope are below 13°, with an exception of two cases in New Zealand (Weiler et al., 2003) and Oregon (McGuire and McDonnell, 2010) where the catchment slope is about 34°. The $MTT_{ew}$ has been recognized as a functional metric correlated with catchment properties such as topography (McGuire et al., 2005) and soil type (Tetzlaff et al., 2009). The effect of slope on young water fraction was highlighted in Jasechko et al. (2016). Besides, temporal sensitivity analysis of parameters which can diagnose the main controls of model under changing
condition (Herman et al., 2013a) is hypothesized to be variant and dominate $MTT_{ew}$ during a rainstorm. Previous studies may have provided a general picture of global $MTT_{ew}$, more specific event-scale $MTT_{ew}$ in steep landscape with abundant precipitation has not yet been investigated.

In this study, two adjacent steep mountainous mesoscale catchments in subtropical northern Taiwan were selected to test our hypotheses. Streamwater and rainwater of eight typhoon events during 2012-2015 were collected with a 3-hr interval
sampling scheme. The TRANSEP was applied to estimate the mean transit time ($MTT_{ew}$) and fraction of event water ($F_{ew}$). Meanwhile, sensitivity analysis was also applied to determine the change of parameter sensitiveness during a rainstorm. Specifically, we test the following hypotheses: (1) $MTT_{ew}$ is shorter and $F_{ew}$ is fluctuated in steep and fractured landscape than those in gentle catchments; (2) the intensity of precipitation input (pulse) is a major control factor on $MTT_{ew}$ and $F_{ew}$, and (3) the parameter sensitiveness which regulates $MTT_{ew}$ and $F_{ew}$ varies under different hydrograph stages. This unique case study
highlighted the $MTT_{ew}$ characteristics in mesoscale mountainous catchments, which may shed new lights on MTT model development in the future.



## 2 Materials and Methods

### 2.1 Study Site

Our study sites, Ping-Lin (PL, 111 km$^2$) and Da-Lin (DL, 79 km$^2$), are two mesoscale mountainous catchments located in northern Taiwan. The elevation ranges from 182 - 994 m and mean slope varies from 26° and 30° (Fig. 1). Interbedded sandstone and shale rock uplifted by the collision of Euroasian plate and Philippine Sea Plate shapes the steep and fractured landscape (Huang et al., 2017). Inceptisol and Endltisol with depth less than 1.0m cover most of the steep hillslopes, while deeper soil of depth around 1.5m lies mostly in the valley bottom (Zehetner et al., 2008). Evergreen conifer-broadleaf forest accounts for 90% of the catchment area. Anthropogenic disturbance is strictly limited by Taipei Water Management Office since the catchments locate within Taipei Water Source Domain, where Feitsui Reservoir is the main domestic water supply for the entire Taipei metropolis.

The two catchments are characterized by subtropical monsoon climate with mean annual temperature of 20°C and precipitation approx. 4 000mm yr$^{-1}$. Specifically, the mean monthly temperature in January and July are 13°C and 26°C, respectively. Besides, there is no significant seasonality in precipitation, but with different precipitation type. In summer (July-September), precipitation falls mainly with high intensity in the form of typhoon (approximately 64% of daily rainfall exceeds 10 mm), whereas the precipitation in winter brought by the East Asian Monsoon is featured by long duration and low intensity (Lin et al., 2015; Zehetner et al., 2008). Echoing the abundant precipitation, the annual runoff is approximately 3,400 mm and the annual potential evapotranspiration is approximately 1 000 mm.

### 2.2 Hydrometrics and water isotope measurement

#### 2.2.1 Hydrometrics

In the study area, there are 5 rain gauges maintained by Taiwan Central Weather Bureau (CWB) and Water Resources Agency (WRA) and two streamflow gauges maintained by WRA providing hourly data (Fig. 1). The hourly precipitation data were interpolated using inverse distance weighting method (exponent coefficient = 2.0) to represent the rainfall heterogeneity (in Supplementary Material). During 2012-2015, eight typhoons were sampled and the basic information, including total precipitation amount (P), duration (D) and intensity of rainfall (I), maximum 3-h rainfall amount (P$_{max3hr}$), total runoff (Q), peak flow (Q$_{max}$), runoff ratio (Q/P) and antecedent 7-day rainfall (AP$_{7day}$) which were considered as an affecting factor on event water transit time and event water fraction in the literature (Hrachowitz et al., 2010; Klaus and McDonnell, 2013; Segura et al., 2012).

#### 2.2.2 Water Isotope Measurement

The two streamwater sampling sites were set at the same location of streamflow gauges near the catchment outlet. Site PL is at the Ping-Lin Bridge and the other site, DL, at the Da-Lin Bridge (Fig. 1). From 2012 to 2015, during every typhoon invasions water samples were collected at 3-hr (storm 1 to 4) to 2-hr (storm 5 to 6) intervals. The sampling period lasted from the





beginning of rainfall to at least 12 hours after the storm. Sampling during typhoon periods in Taiwan takes incredible labor and time not to mention the danger in the extreme weather. Occasional gaps in the stream water sampling were due to the

inaccessibility caused by landslides. Rainwater was also sampled during typhoon. Four rainwater sampling sites were set within the two catchments, but only the one close to site PL was sampled with the same frequency as the streamwater and the others collected rainwater of the typhoon as a whole. The spatial heterogeneity of rainfall and the rainwater isotopic composition could be referred to Supplementary Material (Fig. S1). All water samples were passed through pre-weighed and pre-combusted 0.7-μm GF/F filters in situ. Filtrates were sealed in centrifuge tubes and stored at 4°C until analysis back at the

lab. $\delta^{18}O$ of all the water samples were measured by PICARRO L2120-i isotopic water analyzer which is a time-based measurement system that uses a laser to quantify spectral features of gas phase molecules in an optical cavity. Isotope signature was compared to VSMOW in ‰ with precision of 0.1‰.

**2.3 Estimation of Mean Transit Time ($MTT_{ew}$) and Event Water Fraction ($F_{ew}$)**

The TRANSEP model, generally used for quantifying the mean transit time and event water fraction (Weiler et al., 2003), was

applied onto our catchments. TRANSEP is an event-scale tracer-aided rainfall-runoff model which incorporates the tracer input into instantaneous unit hydrograph (IUH) so that it can analyze water amount and tracer signal synchronously. Below, we briefly described the model and the details could be found in Weiler et al. (2003).

The TRANSEP model consists of two modules, i.e. the streamflow module and the tracer module. The streamflow module is simply an IUH. First, it estimates the effective rainfall ($P_{eff}$) by a loss function that directly contributes to the storm event

(Jakeman and Hornberger, 1993):

$$P_{eff}(t) = p(t)s(t) \tag{1a}$$

$$s(t) = a_1 p(t) + (1 - a_2^{-1})s(t - \Delta t) \tag{1b}$$

where $p(t)$ and $s(t)$ is gross rainfall amount and wetness condition at $t$ time step. The $s(t)$ depends on the antecedent rainfall and ranges from 0 to 1. The parameter $a_1$ constrains the total amount of effective rainfall which equals to direct runoff, and $a_2$

determines the weighting of the rainfall prior to each time step. The antecedent rainfall condition is set as $s(0) = a_3$. Then, the streamflow at $t$ time step $Q(t)$ is calculated by using a convolution of effective rainfall and runoff transfer function $g(\tau)$:

$$Q(t) = \int_0^t g(\tau)P_{eff}(t - \tau)d\tau \tag{2}$$

The parameters in $g(\tau)$ and $s(t)$ could be calibrated by comparing the observed and simulated streamflow.

In the tracer module, the streamflow ($Q$) is separated into pre-event ($Q_p$) and event ($Q_e$) water components, which could

be estimated via tracer data as below:

$$Q = Q_p + Q_e \tag{3}$$

$$CQ = C_p Q_p + C_e Q_e \tag{4}$$

where $C$, $C_p$, and $C_e$ are tracer (e.g., $\delta^{18}O$) concentration in streamwater, pre-event water and event water, respectively. Derived from Eq. (3) and (4), the tracer concentration in streamwater can be simulated by the following equation:



$C(t) = \frac{Q_e(t)}{Q(t)}\left(C_e(t) - C_p\right) + C_p$ (5)

This equation assumes no fractionation effect in $^{18}O$ from either soil evaporation or canopy interception during storms. Streamflow $Q$ is calculated from Eq. (2). Event water component $Q_e$ is estimated by a convolution of event water input and event water transfer function (i.e., $TTD_{ew}$). Pre-event water concentration $C_p$ is a constant for each event. Event water concentration $C_e$ varies with time but the amount and concentration of rainfall are assumed a uniform value. Specifically, the

event water input is estimated through $P_{eff}$ multiplied by the fraction $f$, which is defined as the proportion of effective rainfall reaching the stream during the event. The fraction $f$ is estimated by the loss function again; therefore, $s$, $a_1$, and $a_2$ in Eq. (1b) are replaced by $f$, $b_1$, and $b_2$. Besides, the initial fraction $f(0)$ is zero for there is no event water in the beginning of the event. The event water component $Q_e$ can therefore be estimated by:

$Q_e(t) = \int_0^t P_{eff}(t - \tau)f(t - \tau)h(\tau)d\tau$ (6)

where $h(\tau)$ is the event water transfer function representing a time-invariant $TTD_{ew}$. While capturing the event water component, we could calculate event water concentration $C_e$ by the ratio of labeled rainwater tracers to event water runoff:

$C_e(t) = \frac{\int_0^t C_r(t-\tau)P_{eff}(t-\tau)f(t-\tau)h(\tau)d\tau}{\int_0^t P_{eff}(t-\tau)f(t-\tau)h(\tau)d\tau}$ (7)

where $C_r$ is the tracer (i.e., $\delta^{18}O$) concentration in rainwater. Unlike other hydrograph separation and transit time methods, which need a predefined weighting relationship (Maloszewski et al., 1992), the advantage of this procedure is that it allows

direct weighting of the input concentrations.

In TRANSEP framework, the selection of runoff transfer function and event water transfer function are critical to obtain the $MTT_{ew}$ and the gamma function (GM) is selected to describe the transfer of water in this study for its flexibility (Kirchner et al., 2000; Soulsby et al., 2015). Note that we illustrated parameters $\alpha$ and $\beta$ in two transfer functions by using footprint ($\alpha_q$ and $\beta_q$ as for runoff transfer function, and $\alpha_e$ and $\beta_e$ as for event water transfer function):

$g(\tau) = h(\tau) = \frac{\tau^{\alpha-1}}{\beta^\alpha \Gamma(\alpha)} e^{-\frac{\tau}{\beta}}$ (8)

After optimization of both streamflow and tracer module, the $MTT_{ew}$ ($= \alpha_e \times \beta_e$) and $F_{ew}$ ($= Q_e/Q$) could be estimated and the details of the parameter descriptions and their corresponding ranges were listed in Table 1.

**2.4 Parameter calibration**

Since the model can simulate both streamflow and tracer, $\delta^{18}O$ in this study, simultaneously, a two-step calibration procedure

was suggested (Lu et al., 2017; Weiler et al., 2003). In order to constrain the accuracy of the simulated hydrograph, we calibrated runoff-related parameter sets ($a_1$, $a_2$, $a_3$, and $\alpha_q$, $\beta_q$ used in runoff transfer function, Table 2) via modified Kling–Gupta efficiency coefficient (KGE) (Kling et al., 2012) as simulation performance. The main advantage of KGE is to consider the correlation ($r$), variability ratio ($V$) and bias ratio ($B$) between simulation and observation within a single performance measure:





$KGE = 1 - \sqrt{(r-1)^2 + (V-1)^2 + (B-1)^2}$ (9)

where $r$ is the linear correlation coefficient between the simulated and the observed values, $V$ is a measure of standard deviation ratio of the simulated to the observed values, and $B$ is the ratio between the mean simulated and mean observed, i.e. the bias. In fact, the KGE measures the Euclidean distance from three dimensions, which means that it can optimize correlation, variability, and bias simultaneously. Moreover, $KGE_Q$ value higher than 0.8 was set as the criteria for selecting parameter sets

in order to restrict the simulated streamflow for hydrograph separation. Similarly, we calibrated the tracer-related parameters ($b_1$, $b_2$, and $\alpha_e$, $\beta_e$ in event-water transfer function, Table 1) by $KGE_C$ as well. The two KGEs for each simulation, one for streamflow and the other for the tracer, were taken into account to retrieve the 'best' parameter sets, Pareto front, for each event (Madsen, 2000). In sum, we generated 500,000 parameter sets which were then introduced to the model to get simulations. These simulations were further evaluated by $KGE_Q$ and $KGE_C$ in Pareto front plot for retrieving the 'best' parameter sets.

**2.5 Sensitivity analysis**

Sensitivity analysis (SA) is widely used to improve the understanding of model operation and key parameters. In this study, we used EET (elementary effect test), a global SA proposed by Morris (1991), to screen out parameters which are negligible on the output variability (Pianosi et al., 2016). Specifically, this method samples parameter sets on grids covering the whole parameter space to identify the sensitiveness of each parameter. This method is a one-at-a-time method, which only perturb

parameter $x_i$ along a grid of size $\Delta_i$ without changing other parameters at one time to create a trajectory (parameter set) within the parameter space. The elementary effect (EE) means the output perturbation between the original and the changed input trajectory. The mean of multiple EEs (Morris's $\mu_i$) is taken as a global sensitivity measure for parameter $x_i$:

$\mu_i = \frac{1}{N}\sum_{j=1}^{N}|EE_i^j|$ (10a)

$EE_i = \frac{f(x_1,...,x_i+\Delta_i,...,x_M) - f(x)}{\Delta_i}$ (10b)

where $f(x)$ represents the function evaluation at the previous point in the trajectory. And the root mean squared error (RMSE) is set to represent the $f(x)$ for incorporating measures of model accuracy (Herman et al., 2013b; Pianosi et al., 2016). For measuring the sensitiveness of streamflow and tracer simulation, we computed the $RMSE_Q$ and $RMSE_C$ as the $f(x)$. The sampling size N was set at 20, as Herman et al. (2013a) did. Because parameters $a_1$, $a_2$, and $a_3$ were constrained by the total amount of effective rainfall and direct runoff first, the parameter sensitivity of other six parameters $b_1$, $b_2$, $\alpha_q$, $\beta_q$, $\alpha_e$, $\beta_e$ were

measured for each event and then identify the key parameters under changing conditions.

**3 Results**

**3.1 Hydrometric and tracer dynamics among typhoons**

The hydrometrics of the 12 catchment-events were shown in Table 2. Briefly, the mean typhoon rainfall (P) in PL and DL were 364 and 460 mm within two days, ranging from 236 to 934 mm. The mean rainfall intensity (I) of all events varied from





5.8 to17.8 mm h$^{-1}$, whereas the averaged maximum 3-h rainfall intensity (P$_{max3hr}$) was approx. 90 mm with a large variation from 48 to 189 mm among events. The mean total streamflow (Q) in PL and DL were 217 and 278 mm, which results in a runoff ratio (Q/P) of approx. 0.60 and the peak flow (Q$_{max}$) ranged from 6.0 to 34.1 mm.

The statistics of water isotopic composition of rain water and stream water were shown in Table 3. Generally, there were over 10 samples for each event, except for event 2 and 4 due to their short duration. Regarding inter-event variation, the

averaged $\delta^{18}$O in precipitation of all catchment-events was -8.6 ‰, with a range of -13.3 to -4.9 ‰, in comparison with -6.9 ‰ with a range of -8.1 to -5.4 ‰ in streamwater $\delta^{18}$O. As for intra-event variation, the mean standard deviation (SE) of $\delta^{18}$O was 2.94 in rainwater and 0.76 in streamwater. In addition, the SE of both rainwater $\delta^{18}$O and streamwater $\delta^{18}$O in large events (event 1, 2 and 6) were larger than those in small events (event 3, 4 and 5).

The chrono-sequent variation of rainfall-runoff amount and $\delta^{18}$O of event 5 and 6 in PL were illustrated in Fig. 2 as

examples. Event PL05 and PL06 were selected for having the similar duration; yet, in contrast, the rainfall amount of PL06 was much larger than that of PL05 (Table 2). Thus, the max. 3-hr rainfall intensity of PL06 was 1.6-fold larger than that of PL05. Corresponding to the difference in rainfall, the streamflow amount and peak flow in PL06 were also 2.0-fold and 3.6-fold larger than in PL05. As for water isotopic composition, the $\delta^{18}$O variation of rainwater also positively correlated to rainfall amount, and the variation of $\delta^{18}$O of rainwater within PL05 was gentle than the PL06. Consequently, there was no obvious

variation of $\delta^{18}$O of streamwater within the PL05, but there was larger fluctuation within the PL06.

### 3.2 Simulation Performances and Estimations of MTT$_{ew}$ and F$_{ew}$

The calibrated parameters ($\alpha_e$, $\beta_e$) and the estimated MTT$_{ew}$ and F$_{ew}$ were listed in Table 4 and KGEs, including the V, B, and r, were listed in Table S2 of supplementary material. In general, the both simulations of streamflow and $\delta^{18}$O were satisfactory for all catchment-events. All KGE$_Q$ for the two catchments were higher than 0.85, and the KGE$_C$ of $\delta^{18}$O simulation were also

satisfactory with ranges of 0.96 to 0.99 and 0.75 to 0.90 in PL and DL, respectively. The details of the individual performance of the three perspectives were referred to supplementary material. The parameter $\alpha_e$ varied from 0.68 to 3.01 in PL and from 0.56 to 2.18 in DL. The parameter $\beta_e$ varied from 1.9 to 8.3 in PL and from 2.5 to 12.0 in DL except event DL05. The MTT$_{ew}$ in PL and DL varied from 3.3 to 6.7 hrs and 2.3 to 10.9 hrs, respectively. The ranges and variations of the parameters and MTT$_{ew}$ between the two catchments were similar and consistent. In terms of F$_{ew}$, the variation in PL and DL varied from 0.14

to 0.56 and 0.2 to 0.8 of the total streamflow, respectively, showing that F$_{ew}$ in DL were larger than in PL.

The best-fit simulations of streamflow, event water and streamwater $\delta^{18}$O of event 5 and 6 in PL catchment were demonstrated in Fig. 3. The effective rainfall was much less than the total rainfall in the beginning period due to interception or retention. Later when the cumulative rainfall and soil moisture escalated, the effective rainfall was approaching the total rainfall. As for the hydrograph, the simulated stream flows were close to observed ones in terms of both amount and timing.

The KGEs of streamflow for event 5 and 6 were 0.94 and 0.97, respectively. The performance of $\delta^{18}$O simulations of event PL05 and PL06 could be up to 0.61 and 0.98, respectively. The simulated hydrographs of event water have synchronous responses to streamflow. The F$_{ew}$ values of event PL05 and PL06 were, respectively, 0.2 and 0.56 of total streamflow and there



was no distinct time lag between the peaks of streamflow and event water flow. Furthermore, the event water fraction during the peak period could account for ~0.5 and 0.9, respectively.

## 3.3 Parameter Sensitivity

The global parameters sensitivity analysis for streamflow and tracer module were illustrated in Fig. 4 using scaled Morris's $\mu$ value. Parameter $\alpha_q$, used for runoff transfer function, was determined as the only sensitive parameter in streamflow module and it appeared to increase with the total rainfall, whereas other parameters were not as sensitive. In the tracer module (Fig. 4c), two parameters, $b_1$ and $b_2$, depicted the separation of the fraction of effective rainfall into event water and the decaying effect, respectively (Table 1), but only $b_1$ was sensitive. Since the $\delta^{18}O$ change in streamflow is governed by rain water volume, the fraction of effective rainfall certainly predominates the tracer module, whereas parameter $b_2$ was less sensitive, because it only controls how much previous effective rainfall would recharge into event water. Parameters $\alpha_q$ and $\alpha_e$ were dominant parameters in the tracer module; however, the scale parameters, $\beta_q$ and $\beta_e$, are insensitive. Besides, the three sensitive parameters ($b_1$, $\alpha_q$ and $\alpha_e$) seemed irrelative to the total rainfall.

## 4 Discussions

### 4.1 Rapid response from rainfall to streamflow during typhoons

Steep and fractured landscape along with extreme rainstorms is favorable for, as expected, rapid rainfall–runoff process in mountainous regions (Fig. 2). In fact, the averaged $\delta^{18}O$ in streamwater is highly correlated with that in rainwater, with $R^2 =$ 0.73. Compared with rainwater, streamwater is assumed to be dampened and lagging behind rainfall in terms of $\delta^{18}O$ variability (Lyon et al., 2008), but in Taiwan streamwater almost synchronizes with rainfall, possibly indicating the limited subsurface storage or quick rainfall-runoff response by shallow subsurface flow. Besides, in our two catchments, the average $MTT_{ew}$ is 5.7hr, ranging from 3.3-6.7 and 2.3-10.9 hr in PL and DL, respectively (Table 4), supporting the interpretation of quick response from rainfall to streamflow. Weiler et al. (2003) applied the TRANSEP model onto a steep catchment with a drainage area of 0.17 km$^2$ and average slope of 34 degrees in New Zealand and found that the $MTT_{ew}$ are approx. 10 hr. McGuire and McDonnell (2010) derived the average $MTT_{ew}$ of 20 hr in a 0.11-km$^2$ catchment with the slope of 37 degrees in Oregon via the same model. Our $MTT_{ew}$ is shorter than the two studies, even though our drainage area is over two- to three-order larger than theirs. Segura et al. (2012) identified a positive relationship between catchment area and $MTT_{ew}$; however, our results indicate the size of catchment area is not a first-order control in $MTT_{ew}$, at least, when extending area to 100 km$^2$. Perhaps, part of the event water that usually travels through subsurface reservoir somehow finds "short cuts" in catchments featured by steep and fractured landscape like Taiwan might explain the conflicting results.

From the perspective of rainfall pulse, we compiled 6 previous studies, which all applied TRANSEP model onto catchments with different environmental settings (see supplementary material, Table S3). The $MTT_{ew}$ ranges from 1.0 – 93.8 hr and the $F_{ew}$ from 0.01 – 0.77 (Fig. 5a and 5b). The $MTT_{ew}$ decreases with the increase of rainfall amount. Notably, it is





considerably more diverse (ranging from 1 – 70 hr) in small rainstorm conditions (<100 mm) than large rainstorm conditions
(>100 mm), in which it ranges from 2 – 11 hr. The $MTT_{ew}$ variability in accordance with rainfall amount in our study could possibly be interpreted that with larger rainfall amount, the catchment wetness is increased quickly, which in turn boosts the hydrological connectivity between hillslope and the stream (Jencso et al., 2009; McGuire and McDonnell, 2010).

Figure 6b depicts the relationship between $F_{ew}$ and rainfall amount, in which the $F_{ew}$ is increasing when rainfall amount is less than 30 mm and when rainfall amount reaches beyond this threshold the maximum $F_{ew}$ stays constant at 0.80. In other
words, this indicates how much rainfall reaches the stream directly has an upper limit, though $F_{ew}$ is quite diverse under the same rainfall amount. In fact, $F_{ew}$ increases with rainfall amount, which is similar to other forested lowland or headwater catchments. For example, Johnson et al. (2007) found that pre-event water dominates stormflow ($F_{ew} = 0.21$) for a range of storms in a forested headwater catchment in Amazon; Schellekens et al. (2004) reported that the fraction of rapid flow was 0.44 and 0.81 during small and larger events, respectively, in a forested lowland in tropical headwater catchment (Puerto Rico).
Rainfall amounts actually increase range of $F_{ew}$, and under the maximum line the variation could be controlled by other factors, like landscape and the antecedent condition, which will be introduced in the following section.

### 4.2 Rainfall inputs control on the variability of $MTT_{ew}$, $F_{ew}$ and $\alpha_e$

In this section, we got deeply into the $MTT_{ew}$, $F_{ew}$ as well as $\alpha_e$ with the rainfall inputs in our catchment. The correlation coefficient matrix of parameters against hydrometric factors were shown in supplementary (Table S4). The rainfall intensity
and antecedent condition ($AP_{7day}$) were selected for discussion, since the rainfall intensity has the highest correlation coefficient and antecedent condition was widely discussed in previous studies (Klaus and McDonnell, 2013; Segura et al., 2012).

### 4.2.1 Rainfall intensity controls on $MTT_{ew}$

The relationship of $MTT_{ew}$ with average rainfall intensity and antecedent condition is presented in Fig. 6. $MTT_{ew}$ exponentially decreases with the increase of rainfall intensity ($R^2 = 0.39$) with marginally statistical significance of $p$-value < 0.06 (Fig. 6a).
Despite the marginal significance, $MTT_{ew}$ is generally controlled by rainfall intensity (McGuire and McDonnell, 2010; Segura et al., 2012). Our study confirmed the relationship between $MTT_{ew}$ and rainfall associated indicators (rainfall amount and intensity) broadly. The $MTT_{ew}$ in this study is generally shorter, likely because the greater rainstorm magnitude in Taiwan activates a rapider response in 1) preferential flow processes (Harpold et al., 2010); 2) shallow subsurface flow in perched water tables (McNamara et al., 2005) and 3) saturation excess overland flow (Lyon et al., 2006). In our fractured catchments,
perhaps the soil water (pre-event water) and infiltrated rainwater (event water) would get well mixed as rainfall input increases. If soil water is flushed multiple times, higher and higher proportion of young water would be yielded and thus squeezes the mean transit time to its limit (McGuire and McDonnell, 2010).





### 4.2.2 Rainfall intensity and antecedent condition control on $F_{ew}$

$F_{ew}$ against rainfall intensity and $AP_{7day}$ shows a clear positive correlation in drier conditions ($AP_{7day}$ less than 30 mm) (Fig.

6b, $R^2 = 0.55$, $p$-value $< 0.01$). This correlation cannot be found in three events with wetter antecedent condition (blue dots in Fig. 6b). The positive correlation can be explained by the 'hydrological connectivity' which argues that strong rainstorms can enhance the connecting tendency of flow paths from the hillslope to the stream (James and Roulet, 2009; Jencso et al., 2009; Segura et al., 2012). For example, having observed five consecutive summer rain events in catchments located in the Castkill Mountains of New York, USA, Brown et al. (1999) reported an increase of event water delivered from throughfall and shallow

subsurface flow, which is directly related to rainfall intensity.

Dissimilarly, when the $AP_{7day}$ is larger than 30 mm (blue dots in Fig. 6b), the relationship between $F_{ew}$ and rainfall intensity becomes perplexed by the pre-event water and no clear relationship like in the drier antecedent condition can be found. Summarized from previous studies, the antecedent wetness-$F_{ew}$ relationship can be divided into two distinct phases. In phase one, the $F_{ew}$ decreases with the antecedent wetness, which is supported by TRANSEP modeling (Johnson et al., 2007;

McGuire and McDonnell, 2010; Munoz-Villers and McDonnell, 2012) and soil water studies by Dewalle et al. (1988) and Hornberger et al. (1994)), in which they concluded that the event water fraction is smaller in wetter antecedent conditions. Hypothetically, during wet periods, connectivity within the hillslope increases as the saturated zone expands and thus groundwater sources (pre-event water) have a greater contribution to streamflow (McGuire and McDonnell, 2010; Munoz-Villers and McDonnell, 2012). In the second phase, in contrast, the $F_{ew}$ increases with the antecedent wetness, which is

supported by an age tracking benchmark model by Kirchner (2019). His results showed that high antecedent wetness increases the partitioning of rainfall that reaches the stream. This could be a possible explanation of the behavior justifiable to our catchment. In our catchments, the perplex results likely imply that the relationship between antecedent wetness and event water fraction involves both of the phases discussed above, though it needs more investigation to identify the role of antecedent condition in $F_{ew}$.

### 4.2.3 Rainfall intensity controls on $\alpha_e$

The relationship between $\alpha_e$ of gamma distribution and rainfall intensity reveals an exponentially negative regression (Fig. 6c, $R^2 = 0.50$, $p$-value $< 0.05$) in both wet and dry antecedent conditions. Generally, larger $\alpha_e$ value can only be found when rainfall intensity is less than ~10.0 mm h$^{-1}$. When higher rainfall intensity exceeds 10.0 mm h$^{-1}$, $\alpha_e$ would decreases below 1.0. Some studies are skeptical about the influence of rainfall intensity on parameter $\alpha$. Birkel et al. (2012), for instance, concluded that

the $\alpha$ tends to inversely vary with catchment wetness condition, that is, wet condition results in smaller $\alpha$ and shorter MTT, vice versa. Therefore, a consensus that the parameter, $\alpha$, can be used to describe the drainage behavior (well mixed or not) (Berghuijs and Kirchner, 2017; Harman, 2015). When a fixed flow path without dispersion, like exhibit plug flow, a large $\alpha$ would be determined. The dispersive transport over fixed path lengths will yield humped shape of time distribution. For a catchment with $\alpha = 1$, a linear or well-mixed reservoir could be identified. As $\alpha < 1$, the catchment is more likely to exhibit





comparably high degrees of nonlinearity (i.e., high initial peaks and longer tails in the TTDs) since they rapidly activate preferential flow paths and quickly route water to catchment outlet. In our cases, with the increases of rainfall intensity, the $\alpha_e$ reduces from 3.0 to 0.5, implying that the drainage behavior varies from plug flow, through well-mixed, to preferential flow as rainfall intensity increases.

### 4.3 Transition of parameter sensitivity during typhoons

Generally, parameter sensitivity provides identifiability information of parameters to understand the internal operation in simulation and it may vary in response to different conditions (e.g. landscape setting and flow regimes). Parameter $\alpha_q$ is the only sensitive parameter in the streamflow module, while $\alpha_q$, $\alpha_e$ and $b_1$ are the sensitive parameters in the tracer module (Fig. 4). The degree of sensitivity would vary among events. Our cases identify that $\alpha_q$ is the most important parameter for both streamflow and tracer module in the steep humid environment. Since $\alpha_q$ regulates the shape of runoff transfer function, it would

not be surprising that it is highly correlated with rainfall input. However, it becomes more complicated within the tracer module which functions to delineate the mixing processes and changing flow paths, and thus two more sensitive parameters are involved.

        More interestingly, since parameter sensitivity varies within different wetness conditions, it would be inferred that parameter sensitivity also varies within an event. Through investigating parameter sensibility, the dynamics of the model and

the underlying processes can be better understood, for example, the timing of dominance of a given parameter or a process (Pianosi et al., 2016). Therefore, the normalized Morris's $\mu$ with 12-h moving windows at hourly step for three parameters, $\alpha_q$, $\alpha_e$ and $b_1$, are evaluated for investigating the intra-event variability in parameter sensitivities (Fig. 7). Event PL05 (small event) and PL06 (large event) with a similar dry antecedent condition are taken as examples. The small event results in small event water fraction and the peak of event water fraction coincides with the streamflow peak. By contrast, the large event leads to a

large event water fraction and the peak of event water fraction comes sooner than the streamflow peak. Parameter $b_1$ in small event is getting sensitive with time, whereas it becomes dormant in the large event. Parameter $\alpha_q$ is not very sensitive in the small event, but it response to streamflow significantly. As for $\alpha_e$, it gets sensitive with streamflow, forms a plateau around the peak flow and then shrinks along the recession in the small event. However, in the large event, the sensitivity of $\alpha_e$ surges to form a plateau before the peak of streamflow, sustains the plateau through the peak flow and then diminishes. The above

comparison signifies: 1) parameter sensitivity is case-dependent, and 2) the chronosequent parameter sensitivity may present a clockwise or counterclockwise hysteresis or linear response. As storm magnitude increases, the counterclockwise hysteresis occurs. Hydrologic processes such as bedrock detention storage hypothesis (McDonnell, 2003), bedrock exfiltration, or lagged contribution due to enhanced connectivity in upslope regions (Lehmann et al., 2007) may explain the observed hysteretic pattern between rainfall and $\alpha_e$ sensitivity.

Collectively, a conceptual diagram, which shows how transit time distribution of event water varies during an event in fractured catchments, is presented (Fig. 8). The variation of $TTD_{ew}$ and water ages in catchment storage is described in three hydrograph stages. 1) The rising limb stage: event water infiltrates into near subsurface to partially replace the pre-event water



stored in the subsurface, which slowly flows to stream and thus yields a damped $TTD_{ew}$. 2) The peak flow stage: event water tends to connect existing effective flowpaths, known as preferential flows, from upslope and riparian areas, resulting in a sharp

$TTD_{ew}$. The event water traveling through preferential flows with high mobility leads to a younger streamwater age. 3) The recession stage: as rainfall diminishes and event water reduces, partial pre-event water could use the effective flowpaths and, consequently, the streamwater age is becoming older and the $TTD_{ew}$ is turning from sharp to damped. In attempting to clarify the function of event water in subsurface, one should either select the storage response function (Botter, 2012; Harman, 2015), or incorporate the rainfall pulse to adjust parameter (Hrachowitz et al., 2010) for further exploring the controlling factors of

$MTT_{ew}$ in the steep and wet landscape. In short, we suggest that parameter sensitivity should be taken into account for improving the parameterization when conducting advanced estimation of $MTT_{ew}$ and $F_{ew}$.

## 5 Conclusion

Mean transit time of event water ($MTT_{ew}$) highly associated with flow paths underground has earned much attention over the past decade, but rarely been studied in steep and fractured catchments where the shift of flow paths during typhoons is relevant

to many biogeochemical processes. Our $MTT_{ew}$ (with drainage area of 79 and 110 km$^2$) varies around 2-11 h, in the same order of magnitude with the $MTT_{ew}$ in small-scale catchments with area of 0.02-8.8 km$^2$, indicating drainage area may not the only single one controlling factor on $MTT_{ew}$ within the area spanning from 0.02-100 km$^2$. We compiled all $MTT_{ew}$ studies and found that the $MTT_{ew}$ would sharply decrease with the increase of rainfall amount around the world. Notably, the $MTT_{ew}$ in the catchments with gentle slope varied diversely, whereas the $MTT_{ew}$ in our steep catchments only changed a little, particularly

in large rainstorms. It is likely indicative the constrain of hydraulic gradient which enforces the water velocity as the subsurface is saturated during typhoons. Besides, our event water only accounts for approx. 40% of the streamflow and its fraction increases with rainfall amount. More complicatedly, $F_{ew}$ increases within rainfall amount < 30 mm, but stays at a consistent value at 0.8 when rainfall amount is larger than 30mm. Focusing on our cases, the $MTT_{ew}$ which exponentially decreases with the increase of rainfall intensity agrees with the hypothesis which argues that as rainstorm magnitude increases, $MTT_{ew}$ would

decrease and $F_{ew}$ increase since large rainstorms activate preferential flow processes and hence enhance hillslope-stream connectivity. Intriguingly, antecedent wetness baffles the positive relation between rainfall intensities and $F_{ew}$, indicating the mixing mechanism of pre-event and event water still remains a puzzle that warrants further investigation.

Inter-event sensitivity analysis for demonstrated that parameters $b_1$, $\alpha_q$, $\alpha_e$ in our case are most sensitive to tracer simulation performance among events. The shape parameter $\alpha_e$ has been seen as a proxy of drainage behavior from plug flow,

through well-mixed, to preferential flow with its decrease. In our study, the $\alpha_e$ reduces from 3.0 to 0.5 as rainfall intensity magnifies suggesting the possibility of time-variant $\alpha_e$. Parameter $b_1$ was getting sensitive at small rainfall stage, whereas parameter $\alpha_q$ echoed the streamflow significantly and increased rapidly as heavy rainfall stage during an event. Intra-event sensitivity analysis revealed that $\alpha_q$ and $\alpha_e$ was getting sensitive followed streamflow, up to plateau and getting dispersed along the recession, but event-dependent. This sensitivity analysis reveals that the parameters are time-variant during an event, which



implies that rainwater traveling to stream involves different flow paths and presenting nonlinear behaviors. Therefore, a modeling framework which incorporates the time-variant parameterization is particularly required for estimating streamflow transit time in steep and fractured catchments.

**Author contribution**

Conceptualization: JCH. Formal analysis: JYL. Investigation: YTS TYL. Supervision: TRP CTL. Validation: CYL. Writing – 380 original draft: JYL. Writing – review & editing: JCH.

**Competing interests**

The authors declare that they have no conflict of interest.

**Acknowledgement**

This study is sponsored by Taiwan Ministry of Science and Technology, (MOST 107-2621-B-002-003-MY3 and MOST 106-385 2116-M-002-020) and NTU Research Center for Future Earth (107L901004). The authors are grateful to Taipei Feitsui Reservoir Administration for providing the hydrological data.

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



**Table 1: TRANSEP model and their uniform sampling ranges for sensitivity analysis and calibration.**

| Parameter | Description | Range |
|---|---|---|
| $a_1$ | Weighting the precipitation | $0.005 - 0.1$ |
| $a_2$ | Exponentially weighting the precipitation backward in time | $1 - 15$ |
| $a_3$ | Initial antecedent precipitation index | $0 - 1$ |
| $\alpha_q$ | shape parameter in runoff transfer function | $0.1 - 3.5$ |
| $\beta_q$ | scale parameter in runoff transfer function | $1 - 100$ |
| $b_1$ | Weighting the $P_{eff}$ | $0 - 100$ |
| $b_2$ | Exponentially weighting the $P_{eff}$ backward in time | $1 - 15$ |
| $\alpha_e$ | shape parameter in event water transfer function | $0.1 - 3.5$ |
| $\beta_e$ | scale parameter in event water transfer function | $1 - 100$ |





**Table 2. Hydrometric characteristics of the catchment-events. P = total precipitation, D = duration and I = intensity of rainfall, $P_{max3hr}$ = maximum 3-h rainfall amount, Q = total streamflow, $Q_{max}$ = peak flow, Q/P = runoff ratio and $AP_{7day}$ = antecedent 7-day rainfall.**

| Catchment Event | Typhoon | Date | P (mm) | D (h) | I (mm h$^{-1}$) | $P_{max3hr}$ (mm) | Q (mm) | $Q_{max}$ (mm) | Q/P (-) | $AP_{7day}$ (mm) |
|---|---|---|---|---|---|---|---|---|---|---|
| PL01 | Saola | 2012/7/31 | 699 | 78 | 9.0 | 116 | 440 | 20.9 | 0.63 | 73 |
| PL02 | Soulik | 2013/7/12 | 253 | 23 | 11.0 | 88 | 120 | 12.5 | 0.47 | 2 |
| PL03 | Trami | 2013/8/21 | 319 | 47 | 6.8 | 88 | 149 | 11.9 | 0.47 | 22 |
| PL04 | Matmo | 2014/7/22 | 236 | 34 | 6.9 | 54 | 135 | 6.7 | 0.57 | 0 |
| PL05 | Chan-hom | 2015/7/9 | 261 | 45 | 5.8 | 48 | 158 | 6.2 | 0.61 | 27 |
| PL06 | Soudelor | 2015/8/7 | 414 | 48 | 8.6 | 107 | 299 | 22.5 | 0.72 | 19 |
|  | Average |  | 364 | 46 | 8.0 | 84 | 217 | 13.5 | 0.58 | 24 |
| DL01 | Saola | 2012/7/31 | 934 | 86 | 10.9 | 121 | 621 | 27.0 | 0.66 | 135 |
| DL02 | Soulik | 2013/7/12 | 374 | 21 | 17.8 | 120 | 196 | 21.4 | 0.52 | 1 |
| DL03 | Trami | 2013/8/21 | 345 | 43 | 8.0 | 92 | 171 | 13.3 | 0.50 | 16 |
| DL04 | Matmo | 2014/7/22 | 249 | 29 | 8.6 | 55 | 178 | 9.2 | 0.71 | 6 |
| DL05 | Chan-hom | 2015/7/9 | 247 | 38 | 6.5 | 46 | 162 | 6.0 | 0.66 | 48 |
| DL06 | Soudelor | 2015/8/7 | 612 | 43 | 14.2 | 189 | 337 | 34.1 | 0.55 | 25 |
|  | Average |  | 460 | 43 | 11.0 | 104 | 278 | 18.5 | 0.60 | 39 |



**Table 3: Statistics of δ¹⁸O isotope composition in rainwater and streamwater used as input data for the modeling.**

| Catchment Event | Typhoon | Rainwater (‰) | | | | Streamwater (‰) | | | |
|---|---|---|---|---|---|---|---|---|---|
| | | n | Average* | SE | Max | Min | n | Average* | SE | Max | Min |
| PL01 | Saola | 21 | -11.6 | 3.13 | -3.4 | -15.7 | 24 | -8.6 | 1.18 | -6.3 | -10.7 |
| PL02 | Soulik | 8 | -8.5 | 4.75 | 2.5 | -10.1 | 13 | -7.1 | 0.99 | -4.8 | -8.0 |
| PL03 | Trami | 11 | -14.1 | 4.18 | -5.7 | -18.1 | 15 | -8.5 | 0.94 | -6.5 | -10.0 |
| PL04 | Matmo | 10 | -10.4 | 3.15 | -6.4 | -16.0 | 13 | -7.1 | 1.00 | -5.1 | -7.8 |
| PL05 | Chan-hom | 16 | -5.3 | 1.76 | -3.2 | -8.3 | 21 | -5.6 | 0.41 | -4.8 | -6.3 |
| PL06 | Soudelor | 13 | -9.3 | 4.12 | -1.0 | -14.1 | 18 | -7.2 | 1.12 | -5.0 | -8.5 |
| DL01 | Saola | 21 | -11.2 | 3.13 | -3.4 | -15.7 | 24 | -8.8 | 0.94 | -7.0 | -11.0 |
| DL02 | Soulik | 8 | -8.0 | 4.75 | 2.5 | -10.1 | 12 | -6.9 | 0.60 | -5.4 | -7.6 |
| DL03 | Trami | 11 | -13.9 | 4.18 | -5.7 | -18.1 | 15 | -8.8 | 0.96 | -6.8 | -10.3 |
| DL04 | Matmo | 10 | -10.7 | 3.15 | -6.4 | -16.0 | 13 | -7.2 | 0.80 | -5.9 | -8.3 |
| DL05 | Chan-hom | 16 | -4.8 | 1.76 | -3.2 | -8.3 | 21 | -6.8 | 0.62 | -6.1 | -8.6 |
| DL06 | Soudelor | 13 | -9.1 | 4.12 | -1.0 | -14.1 | 18 | -7.5 | 0.97 | -5.7 | -8.7 |

*Weighted the isotope compositions averages by the water amounts.





**Table 4: Determined parameters of event water transfer function and MTT$_{ew}$ and simulated F$_{ew}$.**

| Catchment Event | $\alpha_e$ | | $\beta_e$ | | MTT$_{ew}$ | | F$_{ew}$ | |
|---|---|---|---|---|---|---|---|---|
| PL01 | 0.68 | (0.65 - 0.80) | 8.3 | (6.2 - 8.4) | 5.6 | (4.9 - 5.8) | 0.33 | (0.31 - 0.33) |
| PL02 | 0.80 | (0.76 - 0.80) | 5.4 | (5.4 - 5.7) | 4.3 | (4.3 - 4.5) | 0.39 | (0.39 - 0.40) |
| PL03 | 1.45 | (1.25 - 1.46) | 2.3 | (2.3 - 2.8) | 3.3 | (3.3 - 3.5) | 0.14 | (0.13 - 0.14) |
| PL04 | 1.90 | (1.79 - 1.94) | 3.5 | (3.5 - 4.0) | 6.7 | (6.7 - 7.4) | 0.41 | (0.40 - 0.42) |
| PL05 | 3.01 | (2.85 - 3.21) | 1.9 | (1.7 - 2.0) | 5.7 | (5.5 - 5.9) | 0.20 | (0.20 - 0.21) |
| PL06 | 0.78 | (0.72 - 0.79) | 7.8 | (7.5 - 8.4) | 6.1 | (5.5 - 6.2) | 0.56 | (0.55 - 0.57) |
| DL01 | 1.10 | (0.77 - 1.19) | 4.7 | (4.3 - 7.6) | 5.1 | (4.9 - 6.0) | 0.24 | (0.24 - 0.29) |
| DL02 | 0.56 | (0.54 - 0.62) | 4.0 | (3.9 - 4.9) | 2.3 | (2.3 - 2.7) | 0.59 | (0.56 - 0.59) |
| DL03 | 0.72 | (0.72 - 0.81) | 12.0 | (11.8 - 13) | 8.6 | (8.6 - 10.0) | 0.20 | (0.17 - 0.26) |
| DL04 | 2.18 | (1.72 - 2.52) | 2.5 | (1.8 - 11.2) | 5.4 | (4.5 - 22.8) | 0.28 | (0.28 - 0.56) |
| DL05 | 1.74 | (1.63 - 1.77) | 6.3 | (5.2 - 6.3) | 10.9 | (8.5 - 11.1) | 0.61 | (0.56 - 0.61) |
| DL06 | 0.92 | (0.88 - 1.12) | 4.2 | (3.6 - 4.4) | 3.8 | (3.8 - 4.3) | 0.80 | (78.8 - 81.4) |

*Numbers in parentheses indicate the lower and upper limit among the best-performed simulations





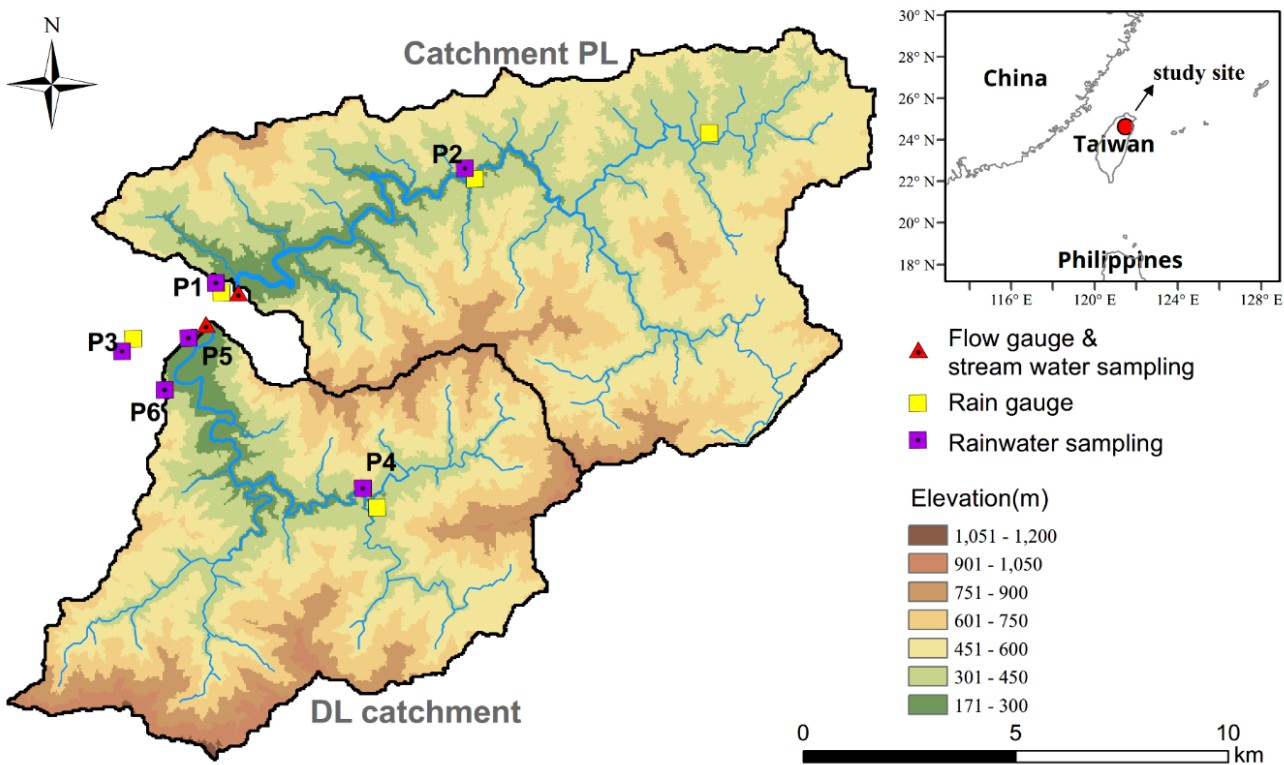

**Figure 1: Location, topographic, and monitoring network of PL and DL catchment. The red triangles are stream water sampling sites and streamflow gauges. The yellow and purple squares represent the location of rain gauges and rainwater sampling sites, respectively.**

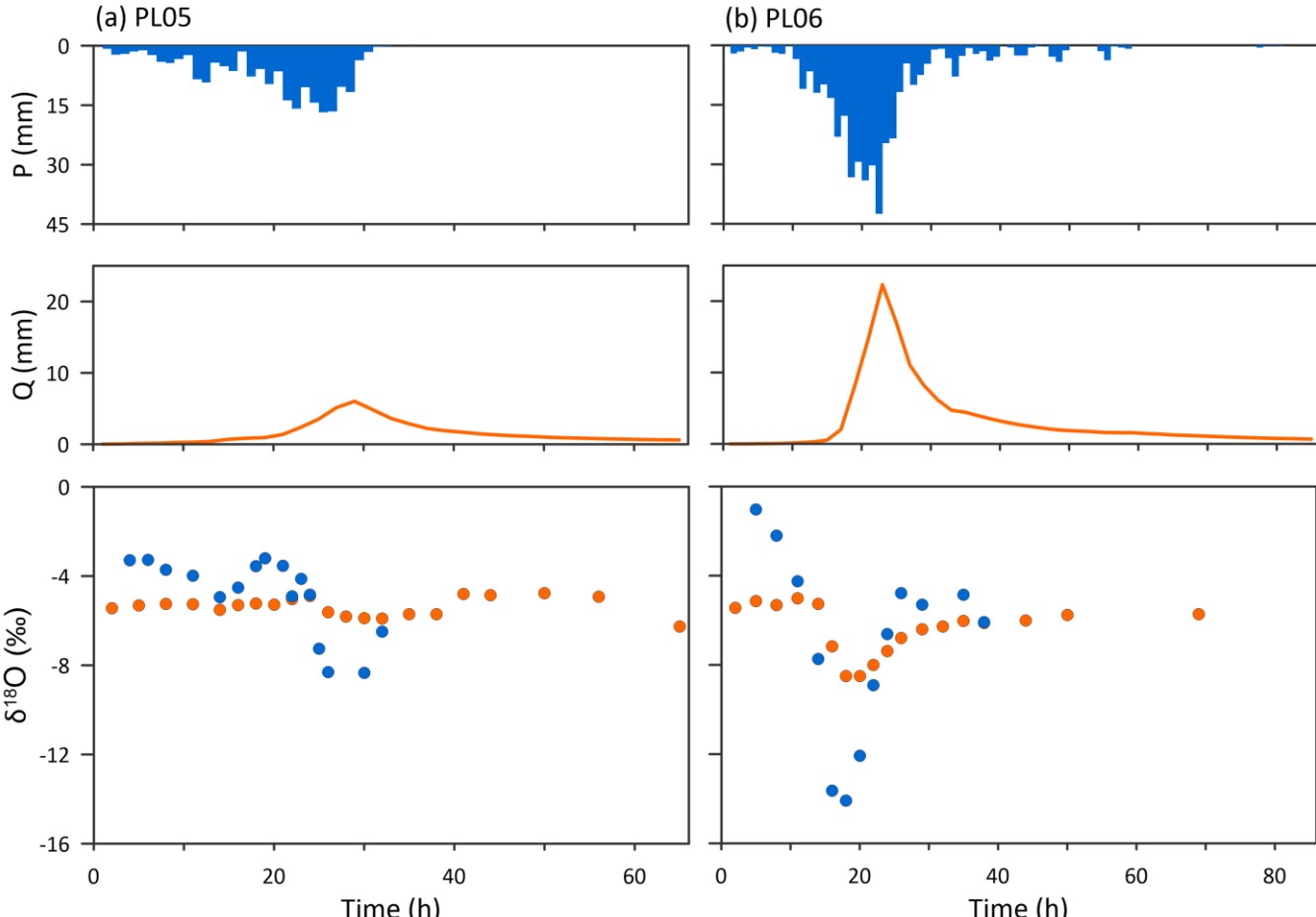

**Figure 2**: **The rainfall (blue bar) and runoff amount (orange line) and their δ¹⁸O (blue and orange dots) during Event PL05 (a) and PL06 (b).**





**Figure 3: Simulations of streamflow and δ18O of PL05 and PL06. The hyetograph and hydrograph of PL05 (a1), (a2) and PL06 (b1), (b2). The observed and simulated δ18O of PL05 (a3) and PL06 (b3).**





Figure 4: Sensitivity metric for parameters in streamflow and tracer module against total rainfall. Total rainfall (a) of the individual catchment-event. Scaled Morris's *µ* value of parameters in streamflow (b) and tracer (c) module, which are derived from their corresponding RMSE.



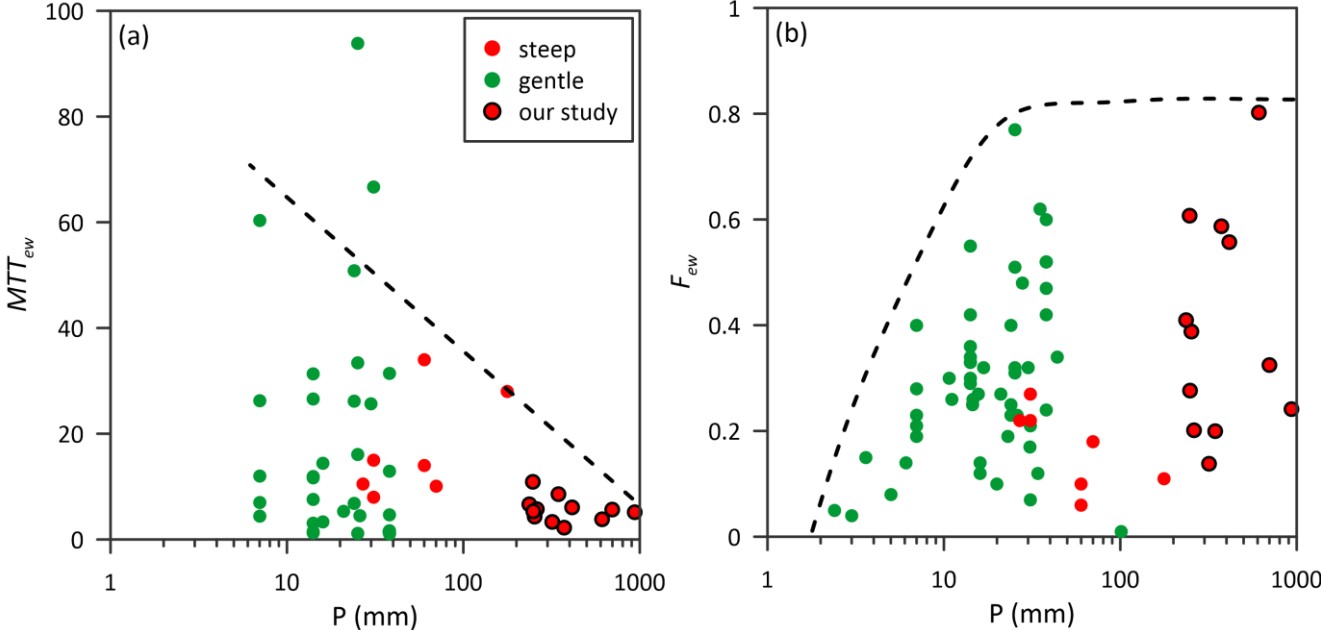

20    **Figure 5: Scatter plots of rainfall against MTT$_{ew}$ (a) and F$_{ew}$ (b) in compiled literature (Table S3) with different catchment gradient. The green, red, and red dots with circle represent gentle, steep, and our study catchments, respectively. The arbitrary dashed line indicates the relationship between rainfall and MTT$_{ew}$ and F$_{ew}$.**





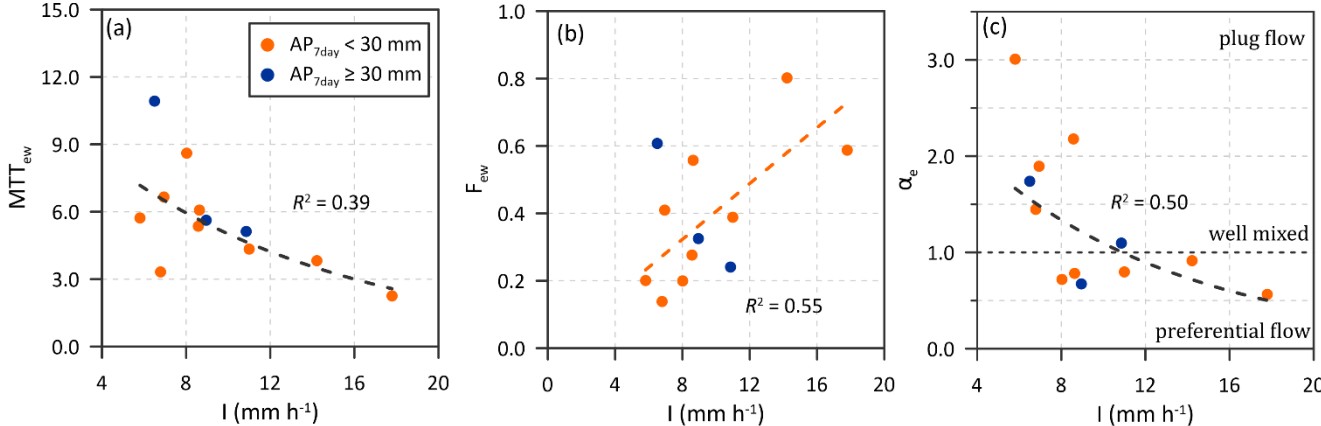

25  **Figure 6: Scatter plots of rainfall input against MTT$_{ew}$ (a), F$_{ew}$ (b), and $\alpha_e$ (c) for all catchment-events. The orange and blue dots represent that antecedent 7-day rainfall (AP$_{7day}$) of 30 mm is set as the criterion for presenting dry and wet condition, respectively. Note that the regression line in plot (b) is only fitted with orange dots. The drainage behaviors corresponding to $\alpha_e$ are labeled on (c).**



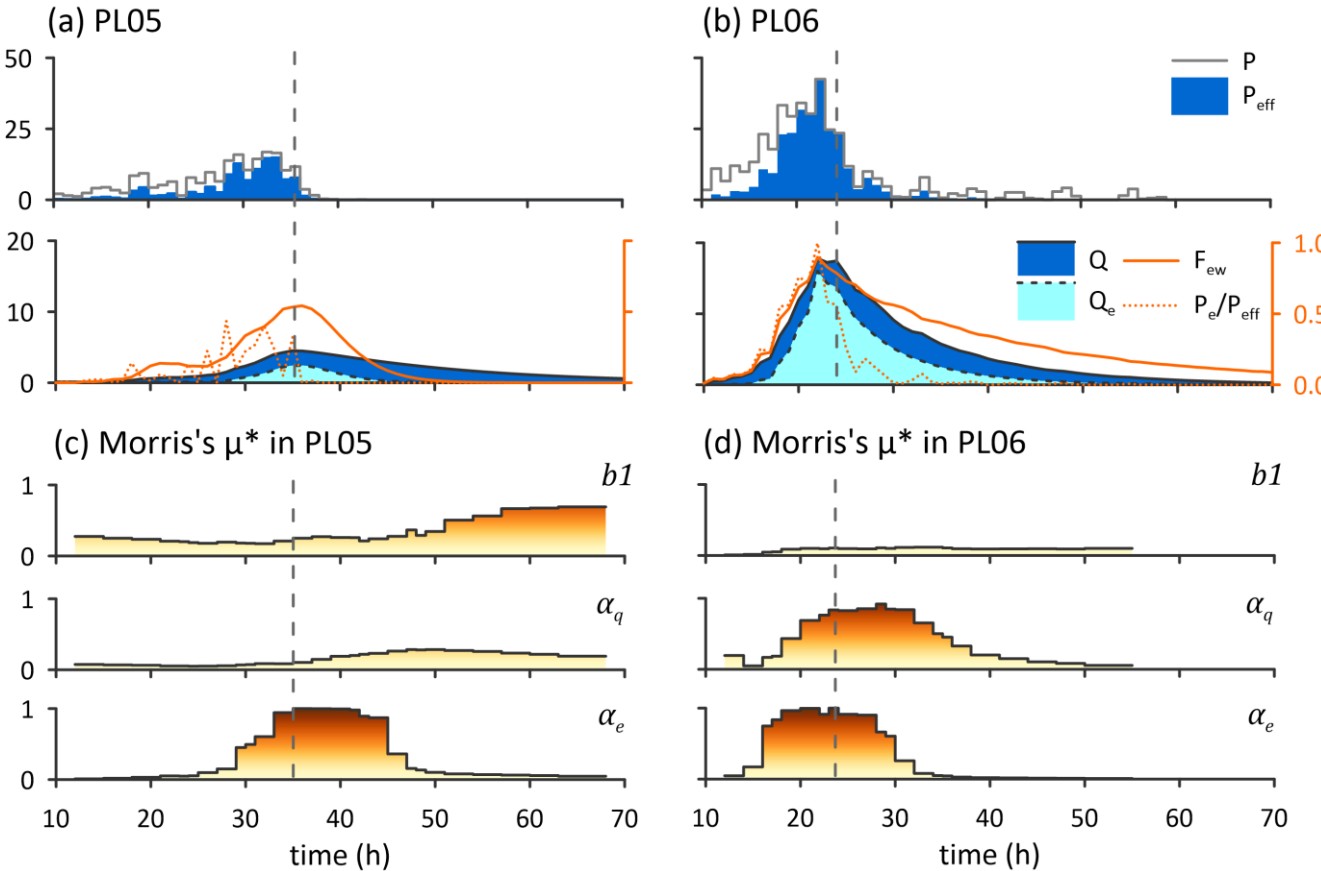

**Figure 7: Time-varying sensitivity calculated with the tracer for three sensitive parameters in Fig. 4. The rainfall – runoff simulation for PL05 (a) and PL06 (b). The scaled sensitivity indices (Morris's μ* value) are calculated for a 12 h moving window with an 1 h time step for PL05 (c) and PL06 (d).**





**Figure 8: The conceptual model of event water movement during typhoon periods with the change of TTDs. The three hydrograph stages are: rising stage (a), peak flow (b) and recession (c). The TTDs demonstrate the behaviors of event water yields corresponding to the three stages.**