# Peer review of "Characterization of event water fractions and transit times under typhoon rainstorms in fractured mountainous catchments: Implications for time-variant parameterization"

_Hydrology and Earth System Sciences, 2019_

## Referee Comment (RC1) · Anonymous Referee #1 · 23 Aug 2019

A transfer-function hydrograph separation model is used to examine event water fraction and transit time of typhoon rainstorms in steep catchments in Taiwan. Few studies of isotope hydrograph separation that involve event water fractions and transit times in high rainfall, subtropical, and steep catchments have been reported in the literature. In addition to a novel catchment setting, a time-variable sensitivity analysis is used to infer processes and controls on event water fractions and characteristics of event water transit time distributions. The results are placed in context with data generated from literature values and show that rainfall intensity influences event water fractions

and characteristics of the transit time distribution. The primary innovation of the study is the application of an existing model to a new setting and the event-dependent and time-variant sensitivity analysis. The study provides additional examples of a model, i.e., a transfer-function hydrograph separation model, that is not widely used, but has a place in the evaluation of event-based isotopic data to improve hydrologic process understanding. The authors conclude the manuscript with a conceptual model that synthesizes hypothesized footpaths and processes across a range of rainfall intensities.

In general, the manuscript makes nice contribution at improving the use and interpretation of transfer-function hydrograph separation models. However, the discussion is fairly one-dimensional (e.g., mostly focused on how rainfall intensity influences event water dynamics) and misses opportunities to connect the time-variable nature of the sensitivity analysis to time-variable transit time studies (e.g., storage selection ideas). In addition, the comparison with catchment structural controls from the literature needs some improvement. For example, little information is provided about the fracture system in the catchments and the authors assume that is part of the reasoning for differences compared with literature values when it could very well be related to soil depth distributions or other aspects of catchment structure. In other words, the conclusions related to bedrock fractures and preferential flow could be strengthened with more evidence that these processes are important over other controls. The authors also compare their transit time findings with those in the literature when some of the literature studies did not examine event water transit time, but rather transit time of total streamflow. This should be sorted out better throughout the manuscript.

The manuscript also requires revision for readability, grammar (plural vs. singular, verb tense, missing "the," etc.), and sentence structure. I found the manuscript not easy to read, but I could follow the general ideas quite well. I did not note all the editorial problems because they are numerous. I suggest a professional editor.

Specific Comments:

Line 26: Suggesting that the transit time "implies information" about something is not quite right. It is the investigator who makes a linkage and implies something about process. Please revise.

Line 29: not only the mean, but other characteristics of the transit time distribution as is discussed in the manuscript (e.g., shape parameters of distributions).

Line 32: Please be clear about what is meant by "event driven pollution." Perhaps give an example.

Line 35: It is not clear to me what the authors mean when they say "tracer-aided rainfall-runoff models." I would also suggest thinking about how the concept of storage selection fits (e.g., Harman 2015).

Line 37: not clear what is widely applied.

Line 41: are lacking in hydroclimatic diversity?

Lines 47-48: This paragraph is convolving the idea of transit time in a general sense (e.g., total stream discharge) vs. event water transit time. For example, I do not believe that McGuire et al. or Tetzlaff et al. ever mention event water. Please keep these as distinct topics.

Line 49: Sentence beginning with "Besides" is unclear.

Line 51: What is a "general picture of global MTT"? Please clarify.

Line 58: Catchments that are compared have comparable mean slope. The evaluation of this hypothesis seems rather weak. Also, other the catchments that are compared have fractured bedrock as well (e.g., the Oregon site, see Gabrielli et al. 2012).

Line 67: Given that one of the hypotheses of this study is related to bedrock fracture flow and the preferential nature of flow paths contributing to streamflow, I would recommend provide additional background/context in the site descriptions. Any information the authors can provide to help provide convincing arguments on controls on the event

water dynamics.

Line 68: Inceptisols and Entisols (spelling) are plural. Also, are the near-stream soils one of these orders? That is not clear.

Lines 97-98: While the supplement describes how spatial variation of the isotopic composition was assessed, it is so critical given the size of the catchments that some of those details should be included in the main text. Further, the assessment provided in the supplement is only based on two storms and from collectors are relatively low elevations. Most of the area of these catchment is above 300 m and altitude effects on rainfall are known to vary by about -0.2 per mil per 100 m. This is should be addressed in the manuscript.

Line 129: This is confusing to me. Why are the amounts and composition assumed a uniform value when you have data on the time variation and the model uses this variation to compute the event water fraction? Does this defeat the purpose of the TRANSEP model?

Line 230: Discussion

Lines 242-43: Can you really claim that catchment area is not "a control" with only two catchments? Your results support previous studies, but your results do not "indicate" that.

Line 270: I cannot find that McGuire and McDonnell examined the correlation between rainfall intensity and MTT_ew. You could say that it is suggested by data presented in McGuire and McDonnell.

Line 277: The term squeezing is not clear.

Line 287: Instead of perplexed, how about "there was no clear relationship?"

Line 300: I think some caution is needed here and throughout the discussion where event water results from this study are compared with other studies when some of the

other studies focused on transit times and distributions that represent all streamflow not just the event water. These distributions are likely very different and not comparable. The discussion also never addresses what is unique about typhoon systems. This seems like one of the main contributions of the study, but it is not made clear in the discussion.

Line 307: What does a "fixed flow path without dispersion" mean?

Line 315: What does a "internal operation in simulation" mean?

Line 324: sensitivity

Line 325: "the timing of dominance of a given parameter or a process" is not clear.

Figure 3: Please include error bounds on isotopic simulations too. Also, be consistent with label and label subplots with letters a through f.

Figure 4: Move y-axis label of sites to the left of the precipitation.

Figure 5: provide slope ranges for gentle and steep.

Figure 7: Label y-axis for subplots a. Also, please add a description of the dashed vertical line in these graphs.

References: Other published work that is relevant and could be added to the literature values in the analysis is by Mosquera et al. (2018).

Gabrielli, C. P., McDonnell, J. J., & Jarvis, W. T. (2012). The role of bedrock groundwater in rainfall–runoff response at hillslope and catchment scales. Journal of Hydrology, 450, 117-133.

Mosquera, G., Segura, C., & Crespo, P. (2018). Flow Partitioning Modelling Using High-Resolution Isotopic and Electrical Conductivity Data. Water, 10(7), 904.

---

## Referee Comment (RC2) · Anonymous Referee #2 · 3 Sep 2019

In the manuscript "Characterization of event water fractions and transit times under typhoon rainstorms in fractured mountainous catchments: Implications for time-variant parameterization" by Lee et al., the authors aim to quantify temporal variability of transit times and event water fractions and to identify controls thereon in a Typhoon-dominated region. They do so by calibrating the TRANSEP model to individual events and then comparing the model parameters and outputs under different conditions. The overall objective of the study is, in principle, worthwhile and the presented data set very interesting. I nevertheless struggle to get enthusiastic about the analysis for several

[Figure]

reasons:

(1) While TRANSEP was a great tool at the time of its development, there has been ample progress in the field of transit time modelling in the 16 years since. Although there is nothing inherently wrong with the use of TRANSEP, it remains elusive to me why the authors chose not to use a simple conceptual model together with the concept of SAS-functions. Calibrating and running such a model, which typically does not have more than 10 parameters, for the entire study period (i.e. also on non-typhoon days) has several advantages. Firstly, the SAS-function formulation directly gives temporally-varying TTDs for each time step as output. From these event fractions can be easily inferred as well. This approach would give a more complete picture of how TTDs are varying throughout the year. In addition, analysis of the model storage dynamics will allow the authors stronger support for many of the interpretations given in the current manuscript, where it is currently essentially speculated that changes in MTT are some-what related to the level of catchment wetness. Secondly, the calibration in such a continuous model would be more robust, as now the 12 (?) parameter model is indi-vidually calibrated against each event, whereby each event only consists of a few data points. Thus, the degree of freedom in the model application unreasonably high. For a continuous model at least the number of stream flow data points to calibrated the model against would considerably increase (while the number of O-18 event samples will remain unchanged).

(2) The first research hypothesis cannot be tested with the available data and the statements made in the conclusion section referring to this hypothesis are thus not supported by the results. Obviously, the authors use results from previous studies to extend their data base and to allow for such an analysis. However, it remains com-pletely unclear which studies these are and how they were chosen. Similarly it remains unclear, which parts of the results discussion and conclusion sections refer to the au-thors own work and which to other work. A much clearer separation is needed here.

(3) The calibration procedure is not described in sufficient detail. It is mentioned that the

best parameter sets were retrieved, based on the two KGE values. How was this done? Per definition, a set of pareto-optimal solutions does not have a single "best" solution. Furthermore, table 4 lists upper and lower limits of parameters. What are these? The set of pareto-optimal solutions? The same question applies to figure 3 – what are the shaded areas around the modelled streamflow? Why are such uncertainty intervals not provided for the O-18 model results in that figure? Why are this uncertainties not considered in figures 5 and 6?

(4) It remains completely unclear which rainfall O-18 data were used in the analysis. Data from 4 sampling locations were available. Were they averaged? Was one chosen?

(5) For most figures and tables: axis and captions need to provide all units and need allow the figure to be standalone. Currently, units are frequently missing and the captions remain unclear.

(6) Figure 2 is redundant with figure 3 and can be removed.

(7) The level of English is rather poor, making large parts of the manuscript difficult to read.

Other points:

p.1,l.11; p.2,l.51: really? What about e.g. Asano and Uchida (2012) or Hale and McDonnell (2016)?

p.2,l.33-35: time-domain convolution and spectral analysis are mathematically essentially equivalent. They cannot be seen as different methods.

p.2,l.35-37: please rephrase – not clear what is meant.

p.2,l.42: "all"? what is meant by that?

p.8,l.237: repetitive – can be removed

p.8,l.242-243: how can you with only 2 stream sampling sites make any statement about the influence of catchment area? What is meant by "extending to 100km2"?

p.8,l.245: which conflicting results?

p.9,l.254: this statement is not warranted! There is nothing that is constant at a value of 0.8 above >30mm (figure 5b). Rather, what can be seen is that there is a lot of variation between ~0.2 and 0.8.

p.9,l.255: the number of data points is too low to call this a "limit"

p.9,l.263: what does this sentence mean?

p.9,l.272-277: likely, but speculative here as not shown by any type of data.

p.10,l.288-299: not sure how this links to this study

p.10,l.302-307: this is very confusing. What do you mean by "sceptical?" The results are all similarly pointing towards an inverse relationship: the higher wetness, the smaller alpha.

References:

Asano, Y., & Uchida, T. (2012). Flow path depth is the main controller of mean base flow transit times in a mountainous catchment. Water Resources Research, 48(3).

Hale, V. C., & McDonnell, J. J. (2016). Effect of bedrock permeability on stream base flow mean transit time scaling relations: 1. A multiscale catchment intercomparison. Water Resources Research, 52(2), 1358-1374.

---

## Author Comment (AC1) · 21 Oct 2019

*A transfer-function hydrograph separation model is used to examine event water fraction and transit time of typhoon rainstorms in steep catchments in Taiwan. Few studies of isotope hydrograph separation that involve event water fractions and transit times in high rainfall, subtropical, and steep catchments have been reported in the literature. In addition to a novel catchment setting, a time-variable sensitivity analysis is used to infer processes and controls on event water fractions and characteristics of event water transit time distributions. The results are placed in context with data generated*

*from literature values and show that rainfall intensity influences event water fractions and characteristics of the transit time distribution. The primary innovation of the study is the application of an existing model to a new setting and the event-dependent and time-variant sensitivity analysis. The study provides additional examples of a model, i.e., a transfer-function hydrograph separation model, that is not widely used, but has a place in the evaluation of event-based isotopic data to improve hydrologic process understanding. The authors conclude the manuscript with a conceptual model that synthesizes hypothesized footpaths and processes across a range of rainfall intensities. In general, the manuscript makes nice contribution at improving the use and interpretation of transfer-function hydrograph separation models.*

Response: We appreciate that reviewer 1 who generously shares his/her expertise in this field fully understands our merits and gives constructive comments for this study.

**General comments**

*However, the discussion is fairly one-dimensional (e.g., mostly focused on how rainfall intensity influences event water dynamics) and misses opportunities to connect the time-variable nature of the sensitivity analysis to time-variable transit time studies (e.g., storage selection ideas).*

Response: Thanks for this comment. In this revision, we discussed that rainfall intensity influences not only event water but also pre-event water dynamics. We also elucidated the connection of our sensitivity analysis to the dynamics of both. Specifically, we thoroughly revised the introduction and discussion in accordance with the comments. The unrefined hypothesis (hypothesis 1) has been removed in introduction. The relevant variables (e.g. catchment size, slope, and antecedent wetness) were discussed in the revised section 4-1. More information about fracture system in our catchments were addressed and some conclusions without evident support by our study were eliminated (like hypothesis 1). We connected our sensitivity analysis to time-variable transit time tightly in the revised section 4-3.

*In addition, the comparison with catchment structural controls from the literature needs some improvement. For example, little information is provided about the fracture system in the catchments and the authors assume that is part of the reasoning for differences compared with literature values when it could very well be related to soil depth distributions or other aspects of catchment structure. In other words, the conclusions related to bedrock fractures and preferential flow could be strengthened with more evidence that these processes are important over other controls. The authors also compare their transit time findings with those in the literature when some of the literature studies did not examine event water transit time, but rather transit time of total streamflow.*

Response: We advanced structural controls of both the literature and our sites. With collected three available structure-associated variables (catchment area, catchment average slope, saturated hydraulic conductivity), we were able to compare the catchment controls that influence $MTT_{ew}$ and $F_{ew}$ between various literature. Unfortunately, the relationships from this compilation is not clear. Only a few of them positively correlates to catchment size. We comprehensively discussed it in the revised section 4-1. As for depicting the fractures in our system, we presented the data of the boring logs in Figure 1 and added the descriptions in Line: 75-78. The revisions can also be found in replying the following Specific Comments.

*This should be sorted out better throughout the manuscript. The manuscript also requires revision for readability, grammar (plural vs. singular, verb tense, missing "the," etc.), and sentence structure. I found the manuscript not easy to read, but I could follow the general ideas quite well. I did not note all the editorial problems because they are numerous. I suggest a professional editor.*

Response: We carefully checked the grammar, sentence structure, and typos in this revision and invited a professional editor to refine the language.

**Specific Comments**

*Line 26: Suggesting that the transit time "implies information" about something is not quite right. It is the investigator who makes a linkage and implies something about process. Please revise.*

Response: We replaced the term, "implies information" with a statement "...is a single integrated measure at the outlet, which..." for clarification in Line: 27.

*Line 29: not only the mean, but other characteristics of the transit time distribution as is discussed in the manuscript (e.g., shape parameters of distributions).*

Response: It was rephrased as: "...characteristics of transit time distribution (TTD, e.g., mean, median and shape parameters of distributions)..." to elucidate all characteristics in Line: 30-31.

*Line 32: Please be clear about what is meant by "event driven pollution." Perhaps give an example.*

Response: The paragraph has been rewritten and the term has been eliminated.

*Line 35: It is not clear to me what the authors mean when they say "tracer-aided rainfall-runoff models." I would also suggest thinking about how the concept of storage selection fits (e.g., Harman 2015).*

Response: We revised the introduction thoroughly in accordance with this suggestion and the reference. Now, the sentence is rephrased as: "Typically, TTDs are mainly derived from an assumed functional form (e.g., exponential or gamma distribution) whose time-variant and time-invariant parameters are estimated from passive tracers and hydrological data (McGuire and McDonnell, 2006; Harman, 2015). While the time-variant parameterization with different methodologies is still under development recently (e.g., Harman, 2015; Soulsby et al., 2015), the time-invariant parameterization has been applied across many regions since 1990." Line: 32-36.

*Line 37: not clear what is widely applied.*

Response: As replied above.

*Line 41: are lacking in hydroclimatic diversity?*

Response: The term is improper and thus removed. We want to express that there are many studies associated with mean transit time of event water so we compiled all those data from different landscape and climatic regions in hope to improve the understanding of the controls. This paragraph has been revised in Line: 51-62.

*Lines 47-48: This paragraph is convolving the idea of transit time in a general sense (e.g., total stream discharge) vs. event water transit time. For example, I do not believe that McGuire et al. or Tetzlaff et al. ever mention event water. Please keep these as distinct topics.*

Response: We carefully used the two terms: event water transit time and transit time of total discharge in this revised paragraph. We also checked the usage of terminology throughout the whole manuscript.

*Line 49: Sentence beginning with "Besides" is unclear.*

Response: The whole paragraph has been re-written and the "Besides" has been eliminated.

*Line 51: What is a "general picture of global MTT"? Please clarify.*

Response: The whole paragraph has been re-written and the "general picture of global MTT" has been eliminated.

*Line 58: Catchments that are compared have comparable mean slope. The evaluation of this hypothesis seems rather weak. Also, other the catchments that are compared have fractured bedrock as well (e.g., the Oregon site, see Gabrielli et al. 2012).*

Response: We recognized that some catchments in Oregon and New Zealand are as steep and fractured as our catchments. The original hypothesis 1 may be a step too far as rethinking about is. We re-wrote this paragraph. Now, it reads as: "Specifically,
we aim to investigate: (1) the control of hydrometric input (rainstorm) on $MTT_{ew}$ and $F_{ew}$ in steep catchments, and (2) the variability of TTD-associated parameter sensitiveness during a rainstorm. This study on the event water MTT of extreme rainstorms in mesoscale mountainous catchments as provided here may shed new light on MTT model development in the future." in Line: 66-69.

*Line 67: Given that one of the hypotheses of this study is related to bedrock fracture flow and the preferential nature of flow paths contributing to streamflow, I would recommend provide additional background/context in the site descriptions. Any information the authors can provide to help provide convincing arguments on controls on the event water dynamics.*

Response: Thanks for the suggestion. Reviewer 1 has mentioned this point in the main concern. We addressed the fractured characteristics in relevant sections and revised Figure 1. In this revision, we added more site descriptions associated with fractured lithology. Now it reads as:" Boring logs show that the lithological formations are highly fractured from ridges to streams (Fig. 1b, provided by Taiwan Central Geological Survey)." in Line: 75-76.

*Line 68: Inceptisols and Entisols (spelling) are plural. Also, are the near-stream soils one of these orders? That is not clear.*

Response: Corrected. We re-wrote this sentence as "Inceptisols with depth less than 1.0 m cover most of the steep hillslopes, while Entisols that have high permeability with soil depth around 1.5 m lie mostly in the valley bottom (Zehetner et al., 2008)" in Line: 76-78.

*Lines 97-98: While the supplement describes how spatial variation of the isotopic composition was assessed, it is so critical given the size of the catchments that some of those details should be included in the main text. Further, the assessment provided in the supplement is only based on two storms and from collectors are relatively low elevations. Most of the area of these catchment is above 300 m and altitude effects on*

[Figure]

*rainfall are known to vary by about -0.2 per mil per 100 m. This is should be addressed in the manuscript.*

Response: Thanks for the reminder. We recognized that altitude effects on rainfall is relevant to mountainous catchment study. In our cases, the relief is around 700m which may induce considerable altitude effect in the isotopic composition of rainwater. However, our previous studies showed that tropical cyclones bring heavy rainfall of violent circulation (well-mixed vapor) so overwhelming that cyclone effect (circulation) dominates the spatial and temporal rainfall patterns. Spatial heterogeneity is still distinct, but compared to orogenic rainfall, it is relatively homogeneous. As for the isotopic composition of rainwater, we conducted rainwater sampling in remote and roadless environments as near-real-time as possible. Our results show that during typhoon periods, the isotopic composition of rainwater might not closely follow altitude effect and the difference among the 4 sites are relatively small. Fischer et al. (2017) also reported the absence of altitude effect in a catchment in Switzerland (relief is 600 m), especially in events with changing wind direction. As they suggested, it is the interaction of atmospheric circulation and the topography which results in the lack of altitude effect in the isotopic composition of rainwater. We added the following sentences in the main text to describe the spatial heterogeneity of rainfall and the isotopic composition of rainwater.

The sentences are: "The spatial heterogeneity of rainfall and the isotopic composition of rainwater were examined. Typhoon-induced rainfall is short-lived, intense, and its spatial heterogeneity in meso-scale catchments is not considerable, because the strong circulation induced by typhoon becomes a predominant control on precipitation over landscape effect (Huang et al., 2012). Our investigation of rainwater isotopic composition, though sparse, also showed an indistinct spatial heterogeneity. Details of the investigation could be referred to in Supplementary Material (Fig. S1)." In Line: 106-111.

*Line 129: This is confusing to me. Why are the amounts and composition assumed*

[Figure]

*a uniform value when you have data on the time variation and the model uses this variation to compute the event water fraction? Does this defeat the purpose of the TRANSEP model?*

Response: Sorry for the unclear description. We rephrased it as: "Note that the time-series amount and isotopic composition of rainwater collected at P1 are used for p and Ce" in Line: 141-142.

*Line 230: Discussion*

Response: Corrected.

*Lines 242-43: Can you really claim that catchment area is not "a control" with only two catchments? Your results support previous studies, but your results do not "indicate" that.*

Response: Yes, the reviewer is right. We cannot really claim that catchment area is not a control. What we intended to say is that the effect of rainfall intensity is becoming a more important control over catchment area as rainfall amount increases over 200 mm. We rephrased our statement in this revision.

*Line 270: I cannot find that McGuire and McDonnell examined the correlation between rainfall intensity and MTTew. You could say that it is suggested by data presented in McGuire and McDonnell.*

Response: Agreed. We rephrased the sentence as: "McGuire and McDonnell (2010) studied the runoff dynamics in H.J. Andrew Experimental Forest in Oregon through successive rainstorms and concluded that the subsurface contributing areas extend far upslope during rainstorms, and the reduction of $MTT_{ew}$ was therefore implied." in Line: 283-285.

*Line 277: The term squeezing is not clear.*

Response: We replaced "squeeze" with "reduce" in Line: 261.

*Line 287: Instead of perplexed, how about "there was no clear relationship?"*

Response: Thanks for the rewording.

*Line 300: I think some caution is needed here and throughout the discussion where event water results from this study are compared with other studies when some of the other studies focused on transit times and distributions that represent all streamflow not just the event water. These distributions are likely very different and not comparable. The discussion also never addresses what is unique about typhoon systems. This seems like one of the main contributions of the study, but it is not made clear in the discussion.*

Response: Thanks for bringing up this issue. We know that the MTT in event water, base flow, and the total discharge mean differently. We used the terms with care in this revision. Because the discussion has been revised thoroughly, we do not post the corresponding revision here to save space.

*Line 307: What does a "fixed flow path without dispersion" mean?*

Response: "Fixed flow path" can be misunderstood. We changed the phrase to "When rainfall transports without dispersion" for clarification in Line: 302.

*Line 315: What does a "internal operation in simulation" mean?*

Response: Rephrased to "the interaction of model functions" in Line: 312-313.

*Line 324: sensitivity*

Response: Corrected. The sentence has been rephrased, see below.

*Line 325: "the timing of dominance of a given parameter or a process" is not clear.*

Response: We tried to clarify it in this revision. Now it reads: "Since parameter sensitivity likely varies depending on different wetness conditions, the change of parameter sensitivity within an event should be demonstrated for understanding the dynamics of
the model and the underlying processes, such as the timing of dominance of a given parameter or a process." in Line: 321-323.

*Figure 3: Please include error bounds on isotopic simulations too. Also, be consistent with label and label subplots with letters a through f.*

Response: Error bounds are included on isotopic simulations, but they are quite narrow due to simulations with high similarity. The label has been corrected.

*Figure 4: Move y-axis label of sites to the left of the precipitation.*

Response: Done.

*Figure 5: provide slope ranges for gentle and steep.*

Response: Added. Note that the original Figure 5 was moved to Figure 4 in this revision.

*Figure 7: Label y-axis for subplots a. Also, please add a description of the dashed vertical line in these graphs.*

Response: The y-axis label is added. "The dashed vertical lines represent the timing of flow peak" is added to the caption for clarification. Note that the original Figure 7 was moved to Figure 6.

*References: Other published work that is relevant and could be added to the literature values in the analysis is by Mosquera et al. (2018).*

Response: Thanks for the references. This insightful study provides a way to test the 'reality' instead of only 'reliability' of model structure via one more proxy. Besides, it also shows that model structure should not be presumed, but need assessment for identification.

Gabrielli, C. P., McDonnell, J. J., Jarvis, W. T. (2012). The role of bedrock groundwater in rainfall–runoff response at hillslope and catchment scales. Journal of Hydrology,

450, 117-133.

Mosquera, G., Segura, C., Crespo, P. (2018). Flow Partitioning Modelling Using High-Resolution Isotopic and Electrical Conductivity Data. Water, 10(7), 904.

Fischer, B. M. C., van Meerveld, H.J., Seibert, J. (2017) Spatial variability in the isotopic composition of rainfall in a small headwater catchment and its effect on hydrograph separation. Journal of Hydrology, 547, 755-769.
* * *

---

## Author Comment (AC2) · 21 Oct 2019

We thank reviewer 2 for the insightful comments which focus on model selection and calibration procedure. In this revised version, we will strengthen the manuscript by adding the rationale why we selected TRANSEP and how we calibrated parameters. The following is a point-by-point response to the reviewer's comments. All corresponding changes in the revised manuscript are underlined.

*In the manuscript "Characterization of event water fractions and transit times under*

[Figure]

*typhoon rainstorms in fractured mountainous catchments: Implications for time-variant parameterization" by Lee et al., the authors aim to quantify temporal variability of transit times and event water fractions and to identify controls thereon in a Typhoon-dominated region. They do so by calibrating the TRANSEP model to individual events and then comparing the model parameters and outputs under different conditions. The overall objective of the study is, in principle, worthwhile and the presented data set very interesting. I nevertheless struggle to get enthusiastic about the analysis for several reasons:*

*(1) While TRANSEP was a great tool at the time of its development, there has been ample progress in the field of transit time modelling in the 16 years since. Although there is nothing inherently wrong with the use of TRANSEP, it remains elusive to me why the authors chose not to use a simple conceptual model together with the concept of SAS-functions. Calibrating and running such a model, which typically does not have more than 10 parameters, for the entire study period (i.e. also on non-typhoon days) has several advantages. Firstly, the SAS-function formulation directly gives temporallyvarying TTDs for each time step as output. From these event fractions can be easily inferred as well. This approach would give a more complete picture of how TTDs are varying throughout the year. In addition, analysis of the model storage dynamics will allow the authors stronger support for many of the interpretations given in the current manuscript, where it is currently essentially speculated that changes in MTT are somewhat related to the level of catchment wetness. Secondly, the calibration in such a continuous model would be more robust, as now the 12 (?) parameter model is individually calibrated against each event, whereby each event only consists of a few data points. Thus, the degree of freedom in the model application unreasonably high. For a continuous model at least the number of stream flow data points to calibrated the model against would considerably increase (while the number of O-18 event samples will remain unchanged).*

Response: Many thanks for this professional comment raised by Reviewer 2, who concerned three issues: (1) the reason for choosing the time-invariant TRANSEP as a tool, not time-variant SAS-functions; (2) Why not using a SAS-function to infer the reasons of MTT changes; (3) the reason why calibrating the events individually instead of a long-term continuous period. We replied the comments below. We almost revised the whole introduction and discussion and part of conclusion to clarify our main theme. Because the revision is of great extent, only the main corresponding changes are attached here.

For the first comment, to our knowledge, time-variant models were developed later and got popular rapidly because they can render TTDs temporally, by which some clues to storage dynamics throughout a given period can be inferred. However, our study tried to compile all $MTT_{ew}$ studies associated with event water; among the few studies some of them were done before time-variant models were developed. So far, Jasecho's work explored the insight of young water fraction in different landscapes. They concluded that young streamflow is less prevalent in steeper landscape due to long flowpaths and high permeability due to fractured rocks. In their study, substantial event water (fast runoff) fraction in steep catchments during rainstorms was observed and recognized. The new found territory shows the necessity of understanding of mean transit time ($MTT_{ew}$) and fraction ($F_{ew}$) of event water during rainstorms [Line: 38-42]. In this regard, compiling all $MTT_{ew}$ studies associated with event water to broadly investigate the potential landscape and climatic controls in different regions is essential for further model development. However, the accumulation of time-variant study at event scale is relatively few. Therefore, we used the time-invariant TRANSEP as our modeling work in comparing with the compilation.

Another reason that reviewer 2 promoted SAS-function is that the analysis of the model storage dynamics can support many of the interpretations; for example, changes in MTT are somewhat related to the level of catchment wetness. We agreed that the SAS-functions indeed have some advantages in interpreting the dynamics of rainfall-runoff transfer. Only one question is that most SAS-functions are presumed, which means

that we can apply various SAS-functions from different philosophies onto catchments and events to straightforwardly test the hypothesis (Kirchner, 2019, we cited it in our manuscript). Alternatively, the exploration between the temporal parameter sensitivity during rainstorms can help us developing hypothesis. For example, why the storage behavior changes from plug flow to preferential flow with the increase of rainfall intensity? Such behavior also infers the mixing degree in storage. In fact, our study in the present form provides us some thinking in storage dynamics, particularly for the role of subsurface storage. It seems that the mixing degree and allocation of different water age in subsurface are more prevalent in our steep catchments. We are currently taking a further step into the theory with a second study.

The third comment is about the calibration procedure. Using a long continuous period has the advantages of lowering the freedom effect. However, this kind of calibration procedure would give a universal parameter-set which can simulate the whole period unbiased (depending on the selection of performance measures); nevertheless the rainstorm period would be overlooked due to the relatively fewer observations. Therefore, some studies, particularly for event simulation, calibrated the parameters among individual events (Weiler et al., 2003; Segura et al., 2012, the two were also cited in our ms). Through the calibration for individual event, the corresponding parameter-set of each event can be used for inferring the rainfall-runoff processes. Reviewer's another concern is freedom effect. This issue still exists and is inevitable for the available data during an event. The only thing we can do for reducing the potential freedom effect is to increase the sampling frequency. Simultaneously taking stream discharge and sequential isotopic data into account can more or less reduce the freedom effect.

*(2) The first research hypothesis cannot be tested with the available data and the statements made in the conclusion section referring to this hypothesis are thus not supported by the results. Obviously, the authors use results from previous studies to extend their data base and to allow for such an analysis. However, it remains completely unclear which studies these are and how they were chosen. Similarly it remains*

[Figure]

*unclear, which parts of the results discussion and conclusion sections refer to the authors own work and which to other work. A much clearer separation is needed here.*

Response: As we replied above, the sections of introduction and discussion were totally revised for clarification. As reviewer pointed out, the original hypothesis 1 and the corresponding statements in conclusion were improper and have been removed in this revision. The goals of this study were rephrased as: "Specifically, we aim to investigate: (1) the control of hydrometric input (rainstorm) on $MTT_{ew}$ and $F_{ew}$ in steep catchments, and (2) the variability of TTD-associated parameter sensitiveness during rainstorm. This study on the event water MTT of extreme rainstorms in mesoscale mountainous catchments as provided here may shed new light on MTT model development in the future." in Line: 66-69. Besides, the data sources of the compilation were listed in supplementary Table S3. We addressed it in Line: 255.

*(3) The calibration procedure is not described in sufficient detail. It is mentioned that the best parameter sets were retrieved, based on the two KGE values. How was this done? Per definition, a set of pareto-optimal solutions does not have a single "best" solution. Furthermore, table 4 lists upper and lower limits of parameters. What are these? The set of pareto-optimal solutions? The same question applies to figure 3 – what are the shaded areas around the modelled streamflow? Why are such uncertainty intervals not provided for the O-18 model results in that figure? Why are this uncertainties not considered in figures 5 and 6?*

Response: We rewrote the text of the calibration procedure (Sect 2.4). First, we used KGEQ > 0.8 as the threshold of reasonable streamflow simulation. Second, we selected the parameter sets on the Pareto front (KGEQ vs KGEC) as representative simulations. Only a parameter was taken as the 'best-performed' parameter set that has the highest KGEC on the Pareto front. In Table 4, upper and lower limits of parameters are the set of Pareto front. In Fig. 2 (original Fig. 3), the shaded areas are streamflow (cyan in Fig. 2b and e), event water (gray in Fig. 2b and e), and O-18 (yellow in Fig. 2c and f) of the representative simulations. We provided the uncertainty intervals in yellow

shaded areas, but they are too narrow to be seen. Also we added error bounds in Fig. 4 and 5. The above descriptions were shown in Line: 172-176.

*(4) It remains completely unclear which rainfall O-18 data were used in the analysis. Data from 4 sampling locations were available. Were they averaged? Was one chosen?*

Response: We only applied rainfall O-18 data in P1 onto the modeling work. Samples of the 4 sites were intermittent because of bad traffic condition during typhoons in this remote and roadless catchments. We clarified it in Line: 105-106.

*(5) For most figures and tables: axis and captions need to provide all units and need allow the figure to be standalone. Currently, units are frequently missing and the captions remain unclear.*

Response: We checked all the figures and table and added axis and units wherever needed. Fig. 5 and 6 (original Fig. 6 and Fig. 7) were corrected. Besides, the captions were rephrased so that the figures can be standalone.

*(6) Figure 2 is redundant with figure 3 and can be removed.*

Response: As reviewer suggested, the original Fig. 2 was merged to Fig. 3. The corresponding text is also re-written into the next paragraph.

*(7) The level of English is rather poor, making large parts of the manuscript difficult to read.*

Response: We carefully checked the grammar error, sentence structure, and typos in this revision and invited a professional editor to refine this manuscript for readability.

**Other points:**

*p.1,l.11; p.2,l.51: really? What about e.g. Asano and Uchida (2012) or Hale and McDonnell (2016)?*

Response: Reviewer is right. Some MTT-associated studies were undertaken in steep catchments. Therefore, the opening sentence is rephrased as: "Transit time estimation has rarely been done for violent rainstorms (e.g., typhoon) in steep and fractured mountainous catchments where the range of transit time, potential controlling factors, and the validity of time-invariant parametrization are unclear. Characterized by steep landscape and frequent typhoon invasions, Taiwan provides great opportunities for inquiring into the above questions." [Line: 11-14]

*p.2,l.33-35: time-domain convolution and spectral analysis are mathematically essentially equivalent. They cannot be seen as different methods.*

Response: The original statement is not clear and a little wordy. The sentence now reads as: "Typically, TTDs are mainly derived from an assumed functional form (e.g., exponential or gamma distribution) whose time-variant and time-invariant parameters are estimated from passive tracers and hydrological data (McGuire and McDonnell, 2006; Harman, 2015). While the time-variant parameterization with different methodologies is still under development recently (e.g., Harman, 2015; Soulsby et al., 2015), the time-invariant parameterization has been applied across many regions since 1990" in this revision. [Line: 32-36]

*p.2,l.35-37: please rephrase – not clear what is meant.*

Response: see above.

*p.2,l.42: "all"? what is meant by that?*

Response: Removed.

*p.8,l.237: repetitive – can be removed*

Response: Removed

*p.8,l.242-243: how can you with only 2 stream sampling sites make any statement about the influence of catchment area? What is meant by "extending to 100km2"?*

Response: Reviewer is right. We should not have leapt too far based on only 2 stream sampling sites. The original sentence has been eliminated and the whole paragraph is revised to avoid the arbitrary statement.

*p.8,l.245: which conflicting results?*

Response: Larger catchments tend to have longer travel time because of longer flow path, but the $MTT_{ew}$ in our sites is short. It just shows that the control of catchment size on MTT needs further studies. We rephrased the sentence as: "However, Segura et al. (2012) concluded a positive relationship between catchment area and $MTT_{ew}$ in small gentle catchments in Quebec. While comparing our $MTT_{ew}$ with the above-mentioned studies (Weiler et al., 2003; McGuire and McDonnell, 2010), we found a controversial relationship." for clarification. [Line: 251-253]

*p.9,l.254: this statement is not warranted! There is nothing that is constant at a value of 0.8 above >30mm (figure 5b). Rather, what can be seen is that there is a lot of variation between 0.2 and 0.8.*

Response: Agreed. This paragraph and the original Fig. 5 have been removed. Instead, we used the compilation to demonstrate the relationship between catchment size with $MTT_{ew}$ and $F_{ew}$, respectively. Also, a new Fig. 4 was added. In this revised paragraph, we found that there is no simple and clear relationship between $MTT_{ew}$ with catchment size, slope, and rainstorm size. Also, $F_{ew}$ shows that further works are needed in able to identify the controls on catchment structure and rainstorm characteristics. This identification would be helpful for the upscaling from observed micro-scale studies to unobserved meso- or macro-scale catchments. On the other hand, the unclear relationship with catchment size implies that perhaps catchment structure (e.g., flow-path length, flow-path gradient, and subsurface permeability) might be a more important control than catchment size, not only for MTT of total streamflow (Tetzlaff et al., 2009; Asano and Uchida, 2012; Hale and McDonnell, 2016), but also for $MTT_{ew}$. The new paragraph is in Line: 254-272.

*p.9,l.255: the number of data points is too low to call this a "limit"*

Response: Replied above.

*p.9,l.263: what does this sentence mean?*

Response: Removed.

*p.9,l.272-277: likely, but speculative here as not shown by any type of data.*

Response: The whole discussion is revised completely. The statements without support by our data were removed.

*p.10,l.288-299: not sure how this links to this study*

Response: This paragraph was thoroughly revised to demonstrate the relationship between Few and rainfall intensity under dry and wet conditions. Now it read as: "Brown et al. (1999) argued that an increase of event water delivered from throughfall and shallow subsurface flow is directly related to rainfall intensity. Such an interpretation is also supported by Segura et al. (2012), but it remains unclear under different antecedent wetness. Recently, Kirchner (2019) developed an age tracking benchmark model to show that high antecedent wetness increases the partitioning of rainfall that reaches the stream resulting in high $F_{ew}$. However, Muñoz-Villers and McDonnell, (2012) applied TRANSEP onto tropical montane catchments and found that the event water contribution decreases with an increasing antecedent wetness condition. They argued that if high permeability of soils and lithologic substrate can lead to vertical rainfall percolation and recharge of deeper layers, the rainfall-runoff responses could be dominated by groundwater discharge, rather than shallow lateral pathways. Although our wet antecedent wetness cases (blue dots) also show a decrease of $F_{ew}$ with the increase of antecedent wetness, it would be hasty to jump into a conclusion with only 3 sampled cases. The controversy of $F_{ew}$ responding to antecedent wetness, therefore, still needs further investigations including a thought on the fracture systems in lithologic substrates, like Gabrielli et. al. (2012) suggested." [Line: 288-298]

*p.10,l.302-307: this is very confusing. What do you mean by "sceptical?" The results are all similarly pointing towards an inverse relationship: the higher wetness, the smaller alpha.*

Response: Sorry, it was the sentences that caused the confusion. Actually, what we wanted to say is: "an inverse relationship: the higher wetness, the smaller alpha" as the reviewer pointed out. Now, the sentences are: "When rainfall transports without dispersion, like plug flow, a large $\alpha$ would be determined. For a catchment with $\alpha = 1$, a linear or well-mixed reservoir could be identified. As $\alpha < 1$, the catchment is more likely to exhibit a relatively high degree of nonlinearity (i.e., high initial peaks and longer tails in the TTDs), indicating preferential flow paths are rapidly activated and quickly route water to the catchment outlet." for clarification. [Line: 302-305]